# Comparative whole-genome analyses of articular chondrocytes and skin fibroblasts reveal distinct genome instability landscapes in mesenchymal cell types

Safia Mahabub Sauty [1], Jacqueline Shine [2], Hamed Bostan [3], Jian-Liang Li [3], Piotr A. Mieczkowski [4], Richard F. Loeser [2,5], Brian O. Diekman [2,5,6]*, Dmitry A. Gordenin [1]*

1 Genome Integrity and Structural Biology Laboratory, National Institute of Environmental Health Sciences, Research Triangle Park, Durham, North Carolina, United States of America, 2 Thurston Arthritis Research Center, University of North Carolina School of Medicine, Chapel Hill, North Carolina, United States of America, 3 Integrative Bioinformatics Support Group, Biostatistics and Computational Biology Branch, National Institute of Environmental Health Sciences, Research Triangle Park, Durham, North Carolina, United States of America, 4 Department of Genetics, University of North Carolina School of Medicine, Chapel Hill, North Carolina, United States of America, 5 Division of Rheumatology, Allergy, and Immunology, University of North Carolina School of Medicine, Chapel Hill, North Carolina, United States of America, 6 Lampe Joint Department of Biomedical Engineering, University of North Carolina at Chapel Hill and North Carolina State University, Raleigh, North Carolina, United States of America

* gordenin@niehs.nih.gov (DAG); bdiekman@email.unc.edu (BOD)

## Abstract

DNA damage lesions can result in mutations and genome rearrangements that are associated with cellular aging and diseases. The landscape of somatic mutations in individual tissue and cell types are dictated by their unique physiological states, cellular functions, mutagenic exposures, and efficiency of DNA repair. Articular chondrocytes and skin fibroblasts are two cell types of mesodermal origin with distinct exposure to internal and external sources of DNA damage. While somatic genome instability features of skin fibroblasts have been well detailed, knowledge about mechanisms underlying genome changes in chondrocytes is scarce. Here, we took a whole-genome sequencing approach to evaluate the load, sources, and patterns of genome changes in 18 primary human chondrocyte clones from donors with and without osteoarthritis (OA). Findings in chondrocyte clones largely agreed with a recent study of 100 single-cell sequenced chondrocytes. We compared genome changes in chondrocytes with clonally-expanded human skin fibroblasts sequenced in our previous studies. We demonstrated that skin fibroblasts show a higher burden of somatic mutations, with an increased rate of mutation accumulation per cell division. Motif-centered analyses of mutation catalogues identified only endogenous sources of mutations in chondrocytes, as opposed to skin fibroblasts which also showed a heavy burden of UV-induced mutations. Spontaneous deamination of meCpG and mutagenesis by exposure to small epoxides and $S_N2$ electrophiles

**Data availability statement:** All numerical data and summary statistics underlying graphs are within the manuscript and its Supporting Information files. BAM and MAF files are available under controlled access in dbGaP database (phs001182.V3, https://www.ncbi.nlm.nih.gov/projects/gap/cgi-bin/study.cgi?study_id=phs001182.v3.p1). P-MACD package to perform motif-centered analyses is available on GitHub (https://github.com/NIEHS/P-MACD).

**Funding:** This work was supported by the US National Institutes of Health Intramural Research Program Project Z1AES103266 to D.A.G., National Institutes of Health R01AG081734 to B.O.D. and National Institutes of Health RO1AG044034 to R.F.L. The funders did not play any role in the study design, data collection and analysis, decision to publish, or preparation of the manuscript.

**Competing interests:** The authors have declared that no competing interests exist.

showed higher mutagenic activities in chondrocytes compared to skin fibroblasts. Chondrocytes showed ubiquitous prevalence of indels in homonucleotide runs of ≥5 bases, while skin fibroblasts showed high contributions of UV-associated deletions of ≥5 bp not in repeats. Structural variants in rearrangement hotspots colocalized with human common fragile sites in skin fibroblasts, but not in chondrocytes. Together, our study comprehensively recorded genome instability features in chondrocytes and highlighted the unique mutagenesis landscapes of two mesenchymal cell types.

## Author summary

Genomes of all human cells accumulate somatic mutations over a lifetime due to exposure to various DNA damaging agents, as well as errors in DNA replication and repair. The source, load, and rate of accumulation of these mutations are variable between individual cell types. High burden of somatic mutations can result in genome instability and contribute to chronic diseases. Hence, it is imperative to identify the modulators of somatic mutagenesis during aging and understand the cell-specific stressors. In this work, we sequence and analyze clonally expanded chondrocyte genomes from donors with and without OA, the most common form of joint disease. We report the baseline levels of somatic SNVs, indels, and large rearrangements in chondrocyte clones and analyze single chondrocyte genomes from a recent paper to corroborate the biological findings. We compare the somatic instability features in chondrocytes with previously sequenced skin fibroblasts. We find higher overall mutagenesis burden in skin fibroblasts compared to chondrocytes, with environmentally induced mutagenic lesions only in skin fibroblasts, and higher levels of endogenous mutagenic activities in chondrocytes. Comparison of these two mesenchymal cell types reveal the roles of anatomic locations and physiological functions in determining the genome instability landscapes of individual cell types.

## Introduction

Accumulation of somatic mutations is inevitable during human life. A substantial burden of somatic mutations underlies the inception of several age-related diseases, including cancers [1–5]. Mathematical deconvolution and statistical analyses of somatic mutations in cancer genomes have identified a plethora of age-dependent and age-independent mutational processes operating in human cancers [6,7]. Statistical modelling of mutations in tumors of self-renewing tissues revealed that at least half of the mutations occurred in pre-cancer states [8]. Advances in sequencing technologies have allowed the detection of these low-burden somatic mutations and their causal mutagenic mechanisms in cancer-free normal tissues [9–11]. Somatic mutagenesis has also been associated with non-neoplastic cardiovascular, neurodegenerative, autoimmune and inflammatory

diseases [3–5]. While mutations accumulate relatively linearly with age in different adult stem cells [12], the rate and spectra of mutation accumulation in individual cell types are modulated by their unique physiological stressors, exposure to mutagens, and rates of errors in DNA replication and damage repair. Hence, comparative analyses of somatic mutations in differentiated cells can unveil cell type-specific histories of DNA damage accumulation and repair associated with chronological aging.

Chondrocytes and skin fibroblasts are two mesenchymal cell types that represent two highly contrasting cellular environments. Chondrocytes are found in cartilage and are adapted to isolated, hypoxic environments with high mechanical stress [13]. Chondrocytes rarely divide in adult articular cartilage after reaching skeletal maturity and can initiate senescence in response to stimuli that are present in the aged or injured joints [13,14]. Abnormal mechanical loading, oxidative damage, metabolic stress, and mitochondrial dysfunction result in chondrocyte degeneration and the onset of OA, a progressive age-related musculoskeletal degenerative disease that affects over 500 million people worldwide [14–17]. Previous studies using comet assay have shown presence of DNA damage in chondrocytes with higher levels in older donors and donors with OA [18,19]. *In vitro* experiments on primary human chondrocytes have shown age-associated decline in DNA damage repair efficiency that was rescued by activation of SIRT6 histone deacetylase [20]. A recent study using single-cell sequencing revealed accumulation of mutations associated with clock-like signatures in chondrocytes [21], however, the range of underlying mutagenic mechanisms remain to be elucidated.

Skin fibroblasts, unlike the terminally differentiated chondrocytes, are proliferative cells that maintain the structural integrity of skin and promote wound healing [22]. Skin provides the first line of defense against all environmental mutagens, with ultraviolet (UV) radiation as the most potent DNA damaging agent. Skin fibroblasts deploy an integrated network of error-prone and error-free DNA repair pathways to resolve these damage-induced lesions, including nucleotide excision repair (NER) and base excision repair (BER) pathways which are known to decline in efficiency with aging [23]. Chronic exposure to UV radiation coupled with declining or compromised damage repair efficiency cause increased persistence of unrepaired DNA lesions that result in genome instability and lead to accelerated skin aging and cancer incidence [24,25]. High burden of UV-associated mutation signatures has been agnostically extracted by signature analyses of melanoma genomes [2,6]. Whole-genome analyses of disease-free skin fibroblasts have revealed both environmental and endogenous sources of somatic genome instabilities [26,27].

Whole-genome sequencing of non-cancerous cells with low burden of mutations can be performed on single cells, single-cell derived clonal populations, microbiopsies, or single molecules from non-clonal population of cells [11,28–31]. Each technology has its advantages and limitations and can be leveraged to cross-validate findings across approaches. In this study, we have sequenced 18 single-cell derived chondrocyte clones from donors with and without OA to characterize the somatic mutation landscape of primary human chondrocytes. To validate our findings in chondrocyte clones, we have performed parallel analyses on 100 single chondrocyte cells derived from donors with and without OA and sequenced in a recent study using single-cell multiple displacement amplification (SCMDA) [21]. The loads, patterns, and sources of somatic mutations in chondrocytes with and without OA were largely similar between these two approaches. We compared somatic genome changes identified in chondrocytes to that in 39 clonal lineages of skin fibroblasts sequenced in our previous studies [26,27]. This comparison served as an initial investigation into whether chondrocytes have a distinct repertoire of genome changes as compared to skin fibroblasts. Our analyses revealed somatic mutations from only endogenous sources in chondrocytes, as opposed to skin fibroblasts which also showed environmental sources of mutations. Chondrocytes showed significantly higher activities of two endogenous mutagenic processes, deamination of methylated CpGs and exposure to small epoxides and $S_N2$ electrophiles, compared to skin fibroblasts as revealed by motif-centered analyses of mutation catalogues. Together, our analyses provided insight into the detectable mutagenic mechanisms operating in chondrocytes and examined the mutation profiles of chondrocytes and skin fibroblasts as a function of physiological state, mutagenic exposures, and DNA repair capacity.

## Results

### Study design

In this study, we determined the repertoire of somatic genome changes in 18 clonally expanded chondrocyte cell cultures of nine donors: six donors with end-stage OA undergoing total knee replacements (samples A01-A12), and three cadaveric donors with healthy femur cartilage (samples B01-B06) (S1 Table). This dataset also includes mutation calls of 100 chondrocyte single cell samples from 17 donors with and without OA sequenced in a recent study [21], as well as 39 clonally expanded skin fibroblast samples from hip (hereafter referred as sun-protected) and forearm (hereafter referred as sun-exposed) of 21 disease-free donors sequenced in our previous studies [26,27] (S1 Table).

Chondrocytes sequenced in this study were derived from donors aged 55–86, with two clones per donor (S1 Table). Bulk cells were plated at high density for 3–4 days of recovery without opportunity for substantial proliferation, as for primary human chondrocytes this requires more extended periods of culture adaptation along with low plating density and growth factor supplementation [32]. Chondrocytes were then plated at a density of 200 cells per 21 cm$^2$ dish for clonal expansion as recently described [33]. Single chondrocyte cells were clonally expanded to over 10$^6$ cells to provide sufficient DNA required for sequencing (S1A Fig). Whole genome sequencing was performed for chondrocyte clones at 39X to 104X average coverage per sample (S1 Table). Bulk chondrocytes from all donors were sequenced at similar depth and used as a surrogate of germline for clone-specific variant calling. All variants detected in a clone but absent from the bulk cell culture from the same donor were considered somatic. All somatic variants were further filtered based on allele fraction (S1B Fig). Skin fibroblast clones were derived and sequenced in previous studies using the same methodology, except donor blood was used as germline surrogates [26,27]. Single cell chondrocytes in Ren *et al.* were sequenced after Multiple displacement amplification (MDA), and bulk chondrocytes were used as germline surrogate [21].

We had previously filtered base substitution and indel calls in clonally expanded skin fibroblasts for 45%-55% (heterozygous) and above 90% (homozygous) allele fractions. Allele fractions for most mutations in skin fibroblasts had fallen within distinct peaks between 45% to 55%, indicating heterozygous mutations [26]. While chondrocyte clones had a higher proportion of mutations with allele fractions below 45% (S2 Fig), we applied a stringent filtering criteria of 45%-55% or ≥90% to retain and analyze only clonal mutations. We did not apply any allele fraction filters to single-chondrocyte mutation calls as each library originates from an individual cell, and the deviations in allele fractions of true somatic mutations beyond expected homozygous and heterozygous range are likely to reflect amplification bias in single chondrocyte genomes. The raw counts of mutations detected in chondrocyte clones prior to allele fraction filtering, as well as the expected number of mutations in single cells after sensitivity correction, are reported in S2f Table in S2 Table along with the final mutation counts for both groups used for all subsequent analyses.

For structural variants in chondrocyte and skin fibroblast clones, calls with 40%-60% (heterozygous) or ≥90% (homozygous) of reads supporting the novel junctions in clones and no reads supporting the junctions in germline surrogates were considered somatic and clonal for deletions, inversions, insertions, and translocations. Duplications were considered somatic and clonal with 20%-40% reads (heterozygous) or 40%-60% reads (homozygous) supporting the junctions in clones and no supporting reads in germline surrogates. We did not attempt to call structural variants in chondrocyte single-cell samples as whole-genome amplification introduces coverage nonuniformity and allelic dropout, making structural-variant detection highly unreliable.

### Mutation loads and accumulation rates in chondrocytes are lower compared to skin fibroblasts

We detected a total of 7,906 clonal base substitutions in the mutation catalogues of 18 clonally expanded chondrocyte genomes (S2a Table in S2 Table). All further analyses were performed with average counts of a specific genome change type(s) across all chondrocyte samples from the same donor. These average counts were calculated from an average of 2 chondrocyte clones per donor for samples sequenced in this study, from 4-10 single chondrocyte cells per donor

sequenced in [21], or, where indicated, separately for subgroups of single chondrocyte OA lesion and OA non-lesion cells from the same donor. We did not see any statistically significant difference in the average base substitution loads between chondrocyte clones from donors with and without OA (healthy donor mean: 447, OA donor mean: 435) (S2b Table in S2 Table). The average substitution loads in chondrocyte clone donors were comparable to that detected in donors of single-cell sequenced chondrocytes, which showed a higher substitution load in healthy donors (healthy donor mean: 521, OA donor mean: 376, p=0.0274, Mann-Whitney) (Fig 1A and S3A Fig and S2b Table in S2 Table). To evaluate whether differences in mutation burden between chondrocyte groups were influenced by the varying age distributions of the cohorts, we modeled mutation burden as a function of age and donor identity using a Linear Mixed-Effects (LME) model treating donor as a random effect to account for inter-individual variability. We compared the resulting residuals between chondrocyte healthy and OA groups using a Mann-Whitney U test. Our analysis of raw mutation counts did not reveal a statistically significant difference in age-adjusted burdens between healthy and OA donors in either the clone or single-cell dataset (S3D Fig and S2g Table in S2 Table). This is in contrast to the previously reported high mutation burden in healthy chondrocyte single-cell donors after correcting for SCMDA mutation detection sensitivity, suggesting that the higher burden may depend strongly on mathematical sensitivity correction or be restricted to a specific subpopulation of cells not captured by clonal propagation. We also detected a significantly higher burden of substitutions in skin fibroblasts compared to both clonally expanded and single cell chondrocyte genomes (Fig 1A and S3A Fig and S2b Table in S2 Table). C➔T changes were the most prevalent base substitution type in both chondrocyte and skin fibroblast genomes (Fig 1A).

Since somatic mutations can be accumulated over an individual's lifetime, we sought to determine if there is any correlation between donor age and average donor substitution burden in our samples (S2c Table in S2 Table). We performed Spearman correlation analyses between donor age and average substitution counts per donor, which revealed significant age-dependent accumulation of substitutions in both chondrocyte clone (Spearman's $\rho$=0.69, p value=0.019) and single cell (Spearman's $\rho$=0.42, p value=0.048) donors (Fig 1B). We performed further correlation analyses using subsets of cadaveric/OA clone donors and healthy/OA lesion/OA non-lesion single chondrocyte donors, all of which lacked significant age correlation likely due to reduced statistical power resulting from limited donor numbers (S3B Fig). Skin fibroblast donors did not show an age-dependent accumulation of mutations, possibly due to the episodic nature of UV mutagenesis (Fig 1B).

The frequency of somatic mutation accumulation is closely linked to the rate of cell proliferation. Mutations at the rate of $10^{-6}$ to $10^{-8}$ per base pair at each cell generation are defined by the efficiency of lesion repair or avoidance, mismatch repair, and the proofreading activity of the replicative polymerases [28,34]. To account for the difference in cell division rates of individual chondrocyte and skin fibroblast clones, we calculated the rate of mutation per cell division based on the shortening of telomeres in each skin fibroblast and chondrocyte clone sample (see materials and methods for details) (S2d Table in S2 Table). Mutation rates normalized per cell division were significantly higher in both sun-protected and sun-exposed skin fibroblasts compared to that of chondrocyte clones (S3C Fig). We performed spearman correlation analyses between donor age and estimated telomere length in the cells that formed clones and detected significant positive correlation between donor age and telomere length in chondrocyte clones, while fibroblasts showed no correlation (S3E Fig). This likely reflects survival bias of the high-fitness progenitor cells capable of *in vitro* colony formation, especially for chondrocyte cells that tend to be quiescent. Short, eroded telomeres act as a critical checkpoint barrier that prevents quiescent cells from re-entering the cell cycle, often triggering senescence or apoptosis [35]. This survival bias is further supported by comparable mutation loads in chondrocyte clones from healthy and OA donors (S3A Fig). We further tested if the accumulation of clock-like mutations correlates with replicative history of the progenitor cells. We performed spearman correlation analysis between telomere-derived cell division estimates and the average mutation load associated with spontaneous deamination of meCpG (C➔T mutations in nCg motif, see later sections for explanation) (S3F Fig). We found no significant correlation between cell division rates and meCpG-associated mutations in either cell type. This independence suggests that mutations in meCpG and replicative attrition of telomeres represent two distinct and decoupled

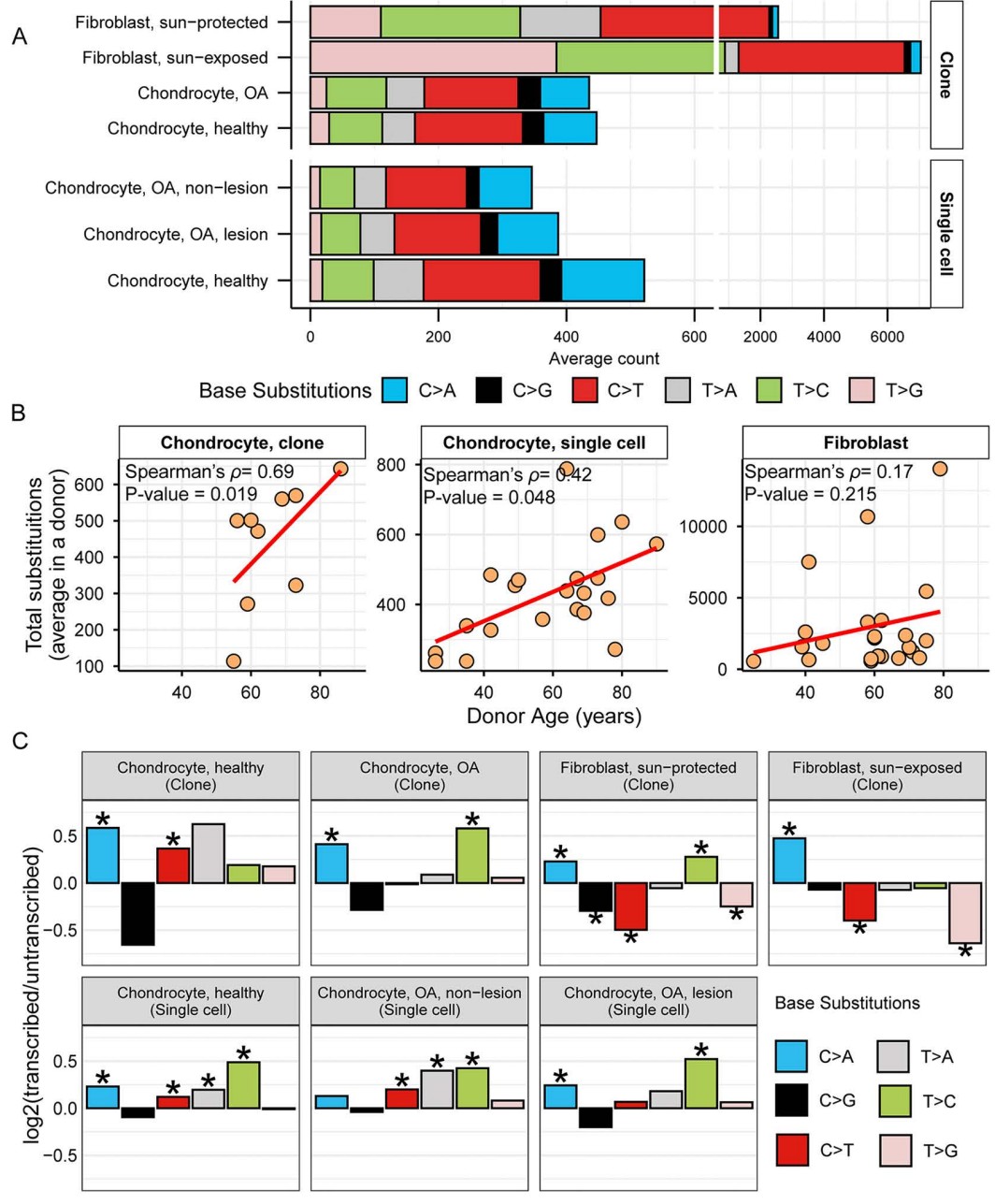

**Fig 1. Spectrum and strand bias of somatic mutations in chondrocytes and skin fibroblasts. A.** Average spectra of base changes including reverse complements detected in chondrocytes and skin fibroblasts, stratified by cell type and sub-group. **B.** Donor mean mutation load plotted against donor age for chondrocyte clones, chondrocyte single-cells, and fibroblast clones. Correlation coefficient and one-sided p value from Spearman's correlation analyses are indicated on each plot. Red line indicates best-fit linear regression. Age correlations within sub-groups of cell types are shown in S3B Fig. Axis ranges for scatter plots in panel B were scaled to the specific range of data points within each cell type in order to maximize the resolution of internal trends and data distributions. For direct quantitative comparisons between groups using standardized scales, refer to the corresponding box plots in S3A Fig. **C.** Ratio between the number of mutations on transcribed and non-transcribed strands detected in each cell type plotted on log2 scale. Asterisk represents two-sided Poisson's test p value <0.05. All source data and p values from statistical analyses are available in S2 Table.

facets reflecting aging of cells that have formed each single-cell colony. Taken together, these data indicate that our single-cell cloning methodology selectively captures a high-fitness cell sub-population.

Next, we annotated the mutations in chondrocyte and skin fibroblast genomes for whether they appear on the transcribed or non-transcribed strand and calculated their relative contributions on each strand (S2e Table in S2 Table). We detected significant enrichment of C➡A mutations on the transcribed strand across all cell types, which may be representative of unrepaired 8-oxo-guanine lesions on the non-transcribed strand [36] (Fig 1C and S2e Table in S2 Table). Significant enrichment of C➡G mutations was detected only on the non-transcribed strand of sun-protected fibroblasts. A possible explanation for this bias could be translesion synthesis carried on by Rev1 across any lesion, including AP-sites created in cytosines on non-transcribed strand [37,38]. Non-transcribed strands of chondrocytes and sun-protected skin fibroblasts also showed a preference for A➡G (reverse complements to T➡C shown as enriched on the transcribed strand in Fig 1C). These changes can originate from mutagenesis by endogenous or exogenous small epoxides and other $S_N2$ reacting electrophiles [39] and references therein; also see next sections). No other base substitution types were enriched on the non-transcribed strands of chondrocyte genomes, as opposed to skin fibroblast genomes which also showed significant enrichments of C➡T and T➡G mutations on the non-transcribed strands (Fig 1C). While C➡T changes represent a major component of UV-mutagenesis, T➡G changes are more difficult to assign to a single mutagenic source. Insertion of cytosine across TT photoproducts by Rev1 is extremely inefficient *in vitro*, however, its structural role in recruiting the REV1-Pol ζ complex is essential for the mutagenic bypass of UV-induced lesions [40,41]. This process may result in sporadic incorporation of a cytosine opposite a damaged thymine, which is subsequently fixed as a T>G transversion during the next round of DNA replication. Importantly, each of the six base substitution types can originate from several kinds of lesions, leading to mutagenesis in the same base. They also can reflect preference of a lesion to transient ssDNA vs DNA-RNA-hybrid formed in R-loops [42]. In summary, we propose that the distinct mutation loads, accumulation rates, and transcription-based asymmetric densities of some base substitution in chondrocytes and skin fibroblasts may be indicative of their unique physiological states, history of mutagenic exposures, and DNA damage repair potentials. Mutational signature and knowledge-based motif-centered analyses presented in the next section aid in more specific linking of base substitution types with environmental and endogenous sources of mutagenesis.

## Motif-centered analyses aid mutational signature analyses in detecting endogenous and environmental sources of mutagenesis in chondrocyte and skin fibroblast genomes

To gain insight into the patterns and sources of mutations in chondrocytes and skin fibroblasts, we first calculated the relative contribution and cosine similarity of COSMIC reference SBS signatures with the 96-trinucleotide motif profiles of different cell sub-groups (S4 Fig). Agnostic signature extraction in our previous study on sun-protected skin fibroblasts showed an overrepresentation of UV-associated signature SBS7b, as well as weak representation of SBS1, SBS2, SBS4, SBS5, SBS11, and SBS18 [26]. Agnostic signature extraction on 100 single chondrocyte genomes reported 3 *de novo* signatures [21]. Signature A showed strong cosine similarity with SBS5, Signature B showed cosine similarity with SBS4, SBS8 and SBS18, and Signature C with SBS42 and SBS44. We generated 96-trinucleotide profiles per chondrocyte and skin fibroblast cell sub-groups and refitted them with published SBS signatures [6]. We detected large contributions of SBS5 (clock-like signature, unknown etiology) and SBS8 (unknown etiology) in all chondrocyte sample groups (Fig 2A and S3a Table in S3 Table). Contributions of SBS1 associated with deamination of meCpG dinucleotides was only detected in chondrocyte clones, but not in single cells (Fig 2A). We detected an entirely distinct set of signatures in skin fibroblast sample groups showing high contributions of UV-associated signatures SBS7a and SBS7b with prominent component of C to T mutations, as well as SBS2 (C to T mutations in APOBEC mutational motifs) which likely stems from its overlap of trinucleotide preference with UV-induced mutations. We detected strong (≥0.8) cosine similarity of SBS5 in OA chondrocyte clones, SBS5 and SBS40 in all single chondrocyte cell sub-groups, and SBS7a in skin fibroblasts (Fig 2B and S3b Table in S3 Table). We note that the consistently detected clock-like signature SBS5 contains all 96 possible mutational

PLOS Genetics

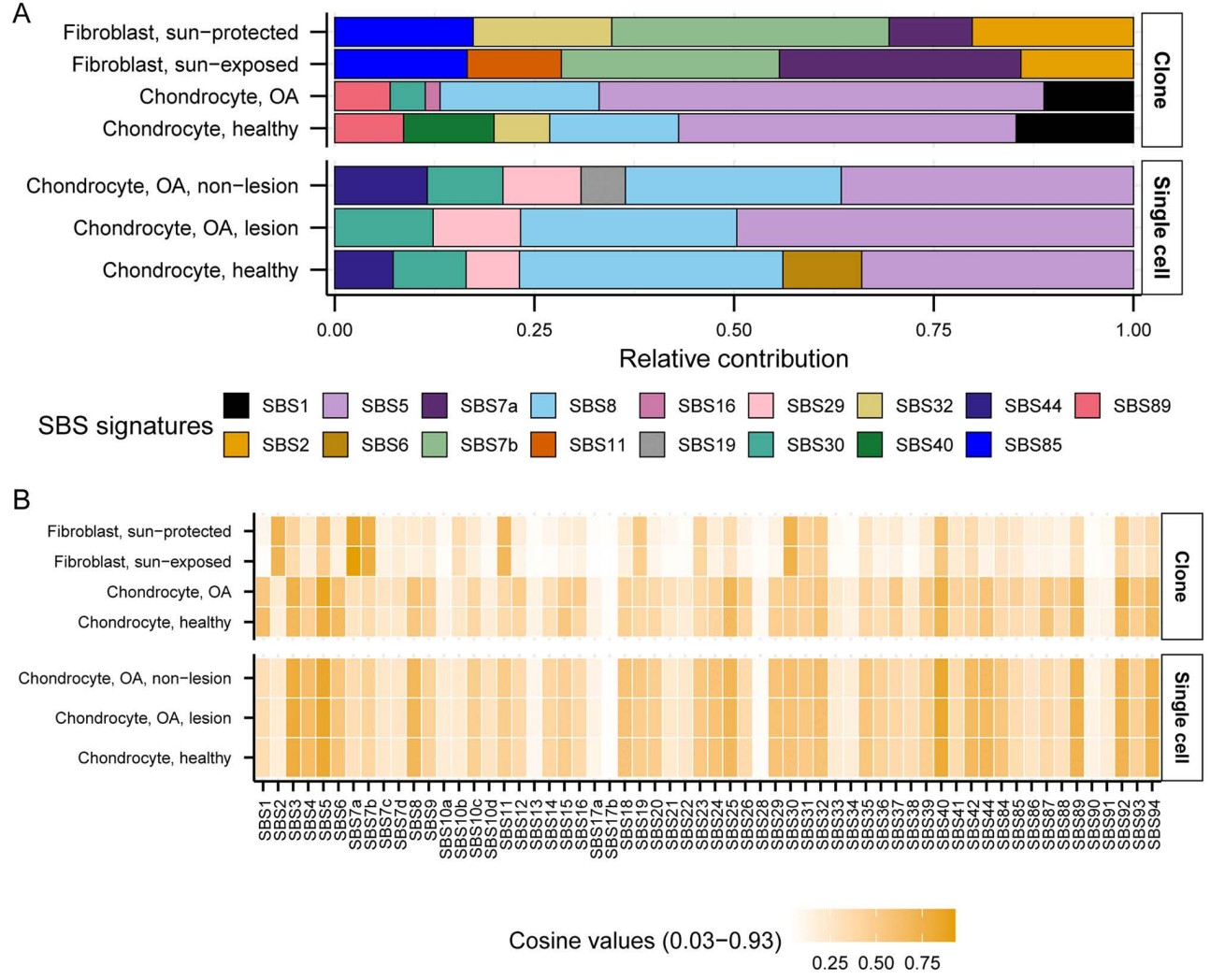

**Fig 2. Relative contribution and cosine similarity of published SBS signatures. A.** 96 trinucleotide profiles were generated from pooled SNVs of indicated cell types and fitted with published COSMIC SBS signatures to determine their relative contributions. **B.** Cosine similarity was calculated between the 96 trinucleotide profiles of indicated cell types and published SBS signatures. All source data are available in S3 Table.

motifs and therefore is likely stemming from several mutagenic sources and mechanisms in all chondrocyte samples (S5 Fig). Furthermore, some signatures like SBS1 and SBS16 have distinct peaks that resemble knowledge-derived mutational motifs but were not detected in all sample groups (Fig 2 and S5 Fig). Therefore, we performed knowledge-based motif-centered analyses to identify the sources of mutations in chondrocytes coming from each donor as previously reported for skin fibroblasts [26,27], cancer catalogues [39,43], and catalogues of somatic mutations in normal tissues [10].

To perform motif-centered analyses, we first curated a list of experimentally validated trinucleotide motifs that are preferentially mutagenized by known mutagenic processes [10] and (S4 Table). We then calculated the enrichment of these mutational motifs in the mutation catalogues of individual samples and donors. Enrichment values are calculated by normalizing all motif associated mutations to all mutations of the same type, all detectable motifs, and all bases in the ± 20 nucleotide genomic context (see Materials and Methods for details). In samples where the enrichment of a mutational

motif is statistically significant, we calculated enrichment-adjusted minimum estimate of mutation load (hereafter referred to as MEML) that can be attributed to the mutagenic process known to preferentially mutagenize the motif in an individual sample or a donor. This approach enables correlation and association assessments between mechanism-driven mutagenesis with biological features such as age or disease status.

We looked for the statistically significant enrichment of 11 experimentally validated motifs in chondrocytes and skin fibroblasts and calculated their enrichment-adjusted MEML values. In contrast to the findings from the signature analyses, we detected nCg➔nTg motif (hereafter referred as nCg) associated with meCpG deamination in 100% of the donors in all chondrocyte and skin fibroblast cohorts (Fig 3 and S5a Table in S5 Table). aTn➔aCn (hereafter referred as aTn) associated with small epoxides and $S_N2$ electrophiles was detected in 87.5%-100% of chondrocyte donors and 50%-76% of skin fibroblast donors (Fig 3 and S5a Table in S5 Table). Ubiquitous presence of UV-associated motifs yCn➔yTn (hereafter referred as yCn) and nTt➔nCt (hereafter referred as nTt) was detected only in skin fibroblast donors. We also performed

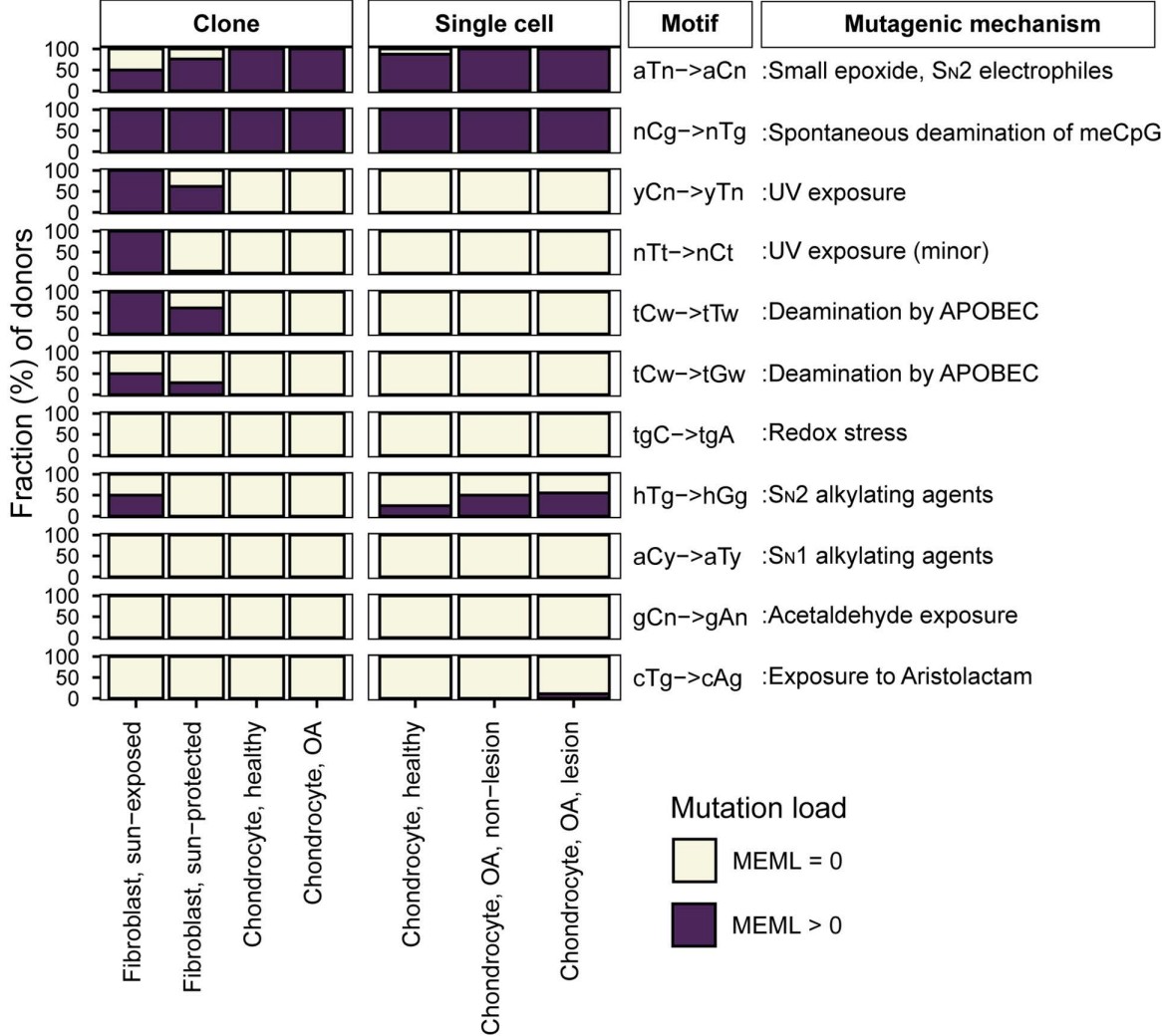

**Fig 3. Prevalence of mutational motifs in chondrocyte and skin fibroblast genomes.** Prevalence of 11 knowledge-based motifs were analyzed, and the percentage of donors with zero and non-zero MEML was plotted for each motif and sub-groups of cell types. All source data are available in S5 Table.

motif-centered analysis with mutations from different allele fractions detected in the chondrocyte clones sequenced in this study. We detected the two ubiquitous motifs aTn and nCg in 100% of donors in low (<45%), stringent (45%-55% or ≥90%), and no (20%-100%) allele fraction filters (S6A Fig and S5a Table in S5 Table). This verified that the low allele fraction mutations in chondrocyte clones are not *in vitro* culture induced, but rather subclonal in nature (i.e., possibly arisen from partial fusion with a neighboring colony). We note that similar to mutational signature analyses, the motif-centered approach can be influenced by confounding mutagenic processes that result in identical base substitutions. However, motif-centered analyses allow interrogation of finer sub-motifs that can help disentangle overlapping mechanisms and more directly assess the activity of specific mutagenic processes. Below, we examine each motif and its associated mechanisms identified in our dataset and compare them across cell types.

### Mutational motif analyses show higher activities of endogenous mutational processes in chondrocytes compared to skin fibroblasts

We first compared the enrichment per donor of two ubiquitous motifs (aTn and nCg), and their sub-motifs, across different chondrocyte sources. We did not detect any statistically significant difference between healthy and OA chondrocyte clones, or among healthy, lesional OA, and non-lesional OA chondrocyte single cells (S6B Fig and S5b Table in S5 Table). We therefore consolidated chondrocytes within each sequencing strategy and compared them with one another as well as with the skin fibroblast sample groups.

aTn motif associated with small epoxides and $S_N2$ electrophiles was identified as a clock-like motif in cancer and disease-free cohorts [10,39]. Exposure to glycidamide, a small epoxide which reacts with nucleobases via $S_N2$ mechanism, caused T to C hypermutation in aTn motif and resembled mutagenic base alkylation preference of $S_N2$ reacting electrophiles in single-stranded DNA (ssDNA) *in vitro* and *in vivo*. Because of its base specificity and ubiquitous nature of the aTn motif in all types of human cancers, we proposed that it is a feature of a broad class of $S_N2$ reacting electrophiles [39] and references therein). We detected a modest but statistically significant increase in aTn enrichment per donor in both chondrocyte clones and single cells compared to that in sun-protected skin fibroblasts (Fig 4A and S5b Table in S5 Table). Enrichment-adjusted average aTn MEML per donor was higher in sun-protected fibroblasts compared to that in chondrocytes (Fig 4B and S5b Table in S5 Table). To account for the significant difference between the total mutation loads detected in chondrocytes and skin fibroblasts, we calculated the contribution of aTn MEML to all AT-pair mutations detected in different cell types. Chondrocytes showed 6%-8% of AT-pair mutations originated from aTn motif, while skin fibroblasts ranged from 0.3% to 3.8% (S7 Fig and S5c Table in S5 Table). T➔C mutations in aTn motif is confounded by T➔C mutations in nTt motif generated by the activity of error-prone trans-lesion synthesis (TLS) across UV-induced lesions [27,44,45]. We analyzed the enrichment and MEML of aTr➔aCr (hereafter referred as aTr) sub-motif which precludes the downstream thymine required for thymine-thymine cyclobutene pyrimidine dimer (CPD) formation and subsequent T➔C mutation caused by UV mutagenesis. Similar to aTn motif, aTr motif showed statistically significant increase in enrichment per donor in chondrocytes compared to skin fibroblasts, while the enrichment-adjusted MEML count was higher in sun-protected skin fibroblasts (Fig 4A and 4B and S5b Table in S5 Table). Spearman correlation analyses of both enrichment per donor and average MEML in a donor between aTn and aTr motifs showed significant positive correlation in all cell types analyzed (Fig 5A and S8A Fig), confirming true small epoxide-associated aTn mutagenesis in both chondrocytes and skin fibroblasts. We also detected a low percentage of sun-exposed skin fibroblast and single cell chondrocyte donors with prevalence of hTg motif associated with exposure to $S_N2$-type alkylating agent Methyl methanesulfonate (MMS) [45], however the number of samples with hTg MEML was very low and potentially representative of stochastic alkylation events in a heterogenous single cell population (Fig 3 and S5g Table in S5 Table). Together, these findings suggest that the mutagenic activities attributable to small-epoxide and broader $S_N2$ electrophile exposure are detectable in both chondrocytes and skin fibroblasts and are more enriched in chondrocytes.

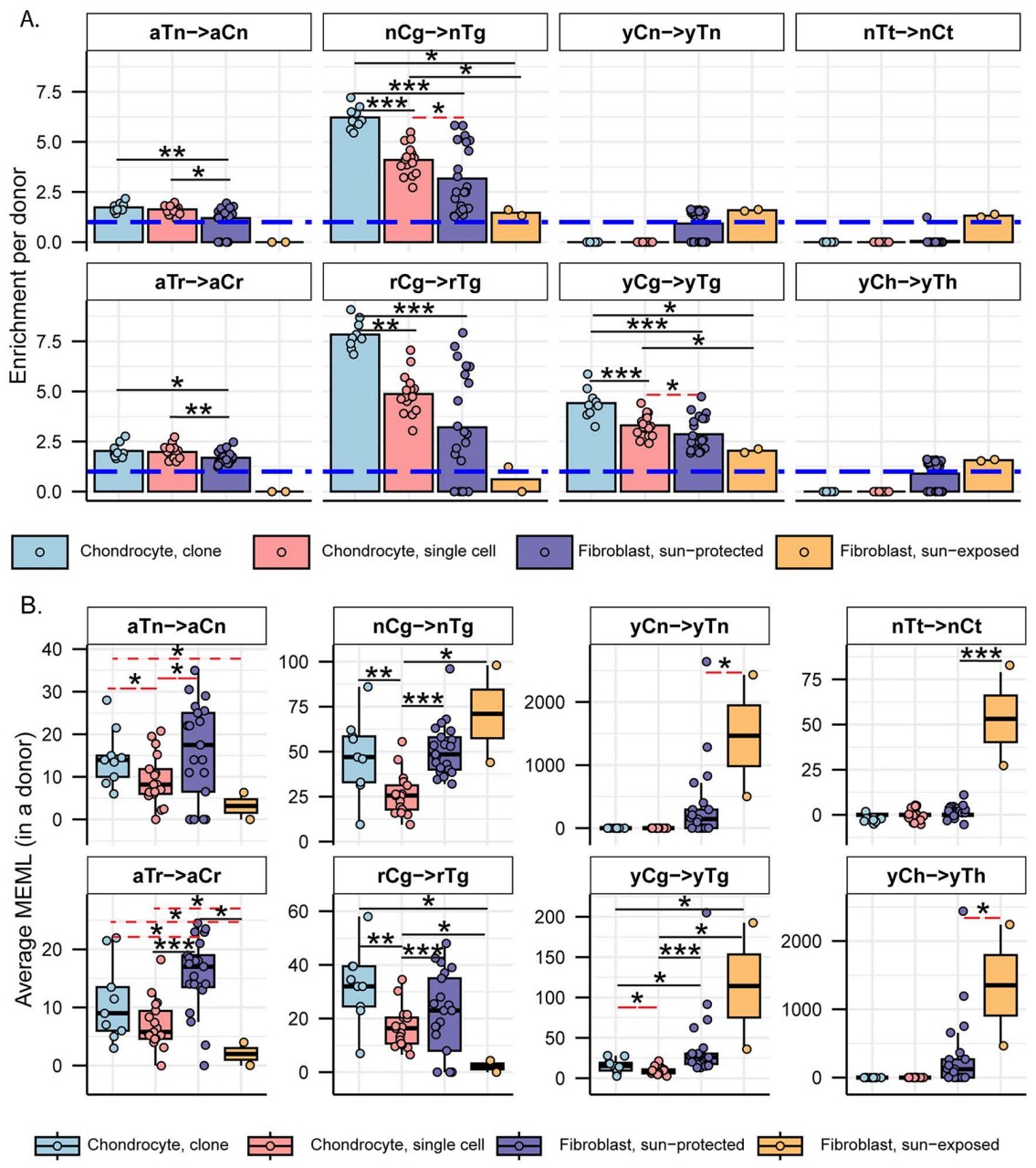

**Fig 4. Analyses of mutational motifs in chondrocyte and skin fibroblast genomes. A.** Fold enrichment of mutational motifs per donor for motifs indicated on panel labels are shown for different cell types. The blue dashed line indicates level of no enrichment. Each dot represents an individual donor. Asterisks represent statistical significance (Wilcoxon Rank Sum test) between connected cell types. Black solid connector lines indicate two-sided test, red dashed connector lines indicate one-sided test. *P value ≤ 0.05, **P value ≤ 0.01, ***P value ≤ 0.001. **B.** The minimum estimate of mutation load (MEML) associated with the mutational motif indicated on panel labels are shown in boxplots for different cell types. Each dot represents average MEML per donor. Asterisks represent statistical significance (Wilcoxon Rank Sum test) between connected cell types. Black solid connector lines indicate two-sided test, red dashed connector lines indicate one-sided test. *P value ≤ 0.05, **P value ≤ 0.01, ***P value ≤ 0.001. All source data and p values from statistical analyses are available in S5 Table.

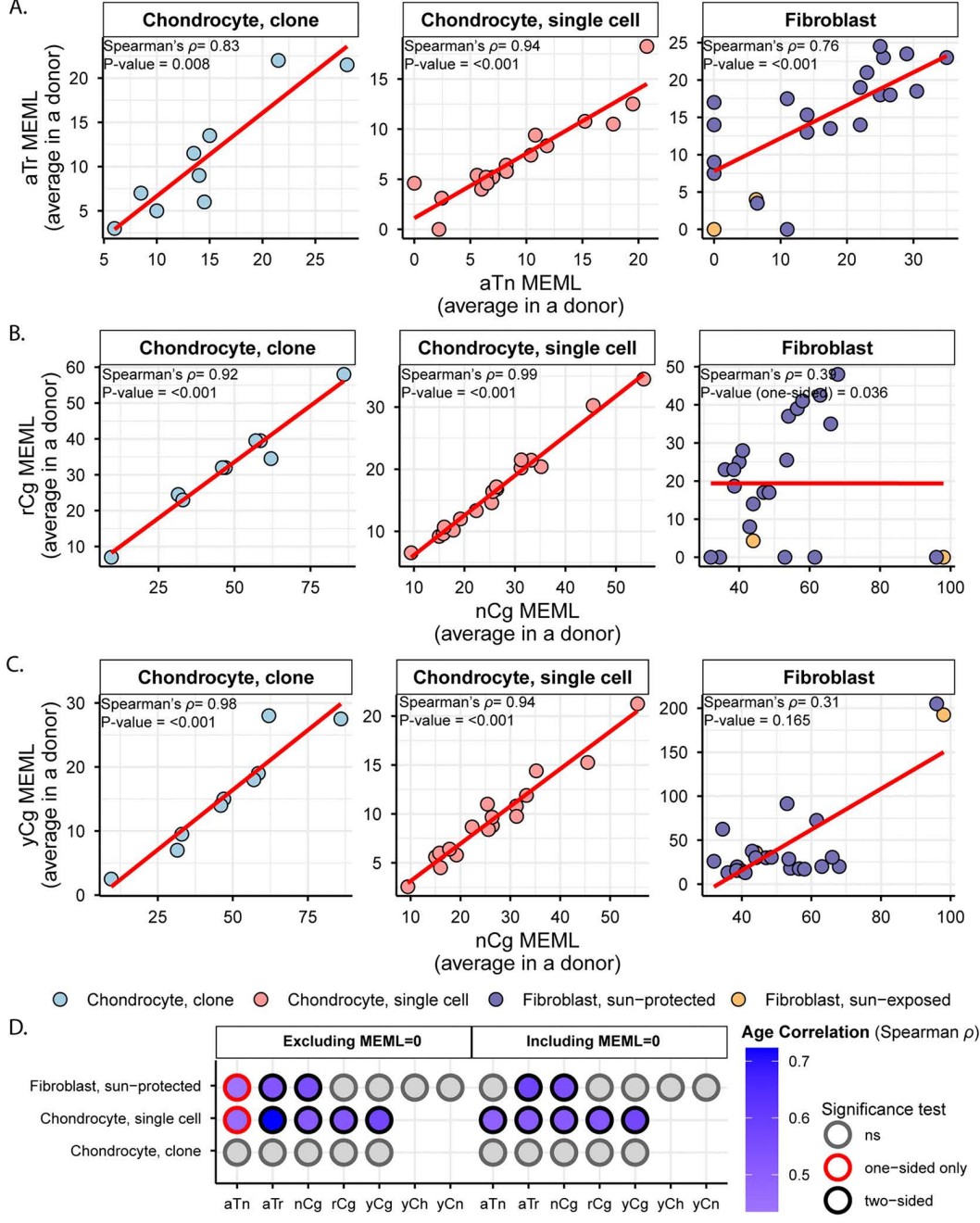

Chondrocyte, clone · Chondrocyte, single cell · Fibroblast, sun−protected · Fibroblast, sun−exposed

**Fig 5. Correlation analyses of aTn and nCg mutational motif MEML with their corresponding sub-motif MEML and donor age. A.** Average aTn MEML per donor are plotted against average aTr MEML per donor for indicated cell types to detect true aTn mutagenesis. Correlation coefficient and two-sided p value from Spearman's correlation analyses are indicated on each plot. Red line indicates best-fit linear regression. Correlation analysis was not conducted for the sun-exposed donors but was still displayed in the plot, as there were only two donors in that group. **B-C.** Average nCg MEML per donor are plotted against **(B)** average rCg MEML per donor and **(C)** average yCg MEML per donor for indicated cell types to detect true nCg mutagenesis. Correlation coefficient and two-sided p value (one-sided for fibroblasts in panel B) from Spearman's correlation analyses are indicated on each plot. Red line indicates best-fit linear regression. Correlation analysis was not conducted for the sun-exposed donors but was still displayed in the plot, as there were only two donors in that group. Axis ranges for scatter plots in panels A-C were scaled to the specific range of data points within each cell type in order to maximize the resolution of internal trends and data distributions. **D.** Spearman's correlation analysis was performed between donor age and average donor MEML excluding (left panel) and including (right panel) zero MEML values. Statistical significance of the correlations is indicated by color. All source data and p values from statistical analyses are available in S5 Table.

Spontaneous deamination of 5-methylcytosines in meCpGs causes C➔T mutations in nCg context [26,45,46]. nCg motif correlates with COSMIC signature SBS1 (S5 Fig) and is detected to be clock-like across many cancer, healthy, and diseased cell types [2,6,10]. We detected a significantly higher nCg enrichment per donor in both chondrocyte clone and single cell donors compared to skin fibroblast donors (Fig 4A and S5b Table in S5 Table). Enrichment adjusted average nCg MEML per donor was comparable between skin fibroblasts and chondrocyte clones and lower in chondrocyte single cells (Fig 4B and S5b Table in S5 Table). Chondrocytes showed higher contributions of nCg MEML to all CG-pair mutations (18% in clones, 9% in single cells) compared to skin fibroblast samples (1% in sun-exposed, 2.5% in sun-protected) (S7 Fig and S5c Table in S5 Table). Multiple mutagenic lesions result in C➔T mutations with overlapping preferred trinucleotide context, thus confounding the assignment of the source of lesions. UV mutagenesis causes C➔T mutations in yCn motif, the bases of which overlap with the bases of meCpG-associated motif nCg. We analyzed two sub-motifs of nCg, rCg (CpG mutation preceded by a purine, does not overlap with UV-associated motif yCn) and yCg (CpG mutation preceded by a pyrimidine, overlaps with UV-associated motif yCn), to detect true meCpG deamination. We detected a significantly higher enrichment per donor of both rCg and yCg sub-motifs in chondrocyte clones and single cells compared to that in skin fibroblasts (Fig 4A and S5b Table in S5 Table). When comparing MEML counts, we found a modest increase in rCg MEML load in chondrocyte clones compared to sun-protected skin fibroblasts (chondrocyte clone mean: 32, sun-protected skin fibroblast mean: 21, two-sided Wilcox test p value 0.11) (Fig 4B and S5b Table in S5 Table). Chondrocyte single cells showed significantly lower rCg MEML compared to chondrocyte clones and sun-protected skin fibroblasts, while sun-exposed skin fibroblasts showed the lowest average rCg MEML count (Fig 4B). This was diametrically opposite to yCg MEML load which was significantly higher in both sun-protected and sun-exposed skin fibroblasts compared to chondrocytes (Fig 4B and S5b Table in S5 Table). We performed motif–motif correlation analyses using both enrichment per donor and the average MEML per donor, comparing nCg with rCg and yCg motifs in all cell types. We detected strong positive correlation of nCg MEML with both rCg MEML and yCg MEML in chondrocytes, as opposed to skin fibroblasts which only showed significant positive correlation with rCg MEML but not yCg MEML potentially confounded by UV-mutagenesis (Fig 5B and 5C and S5d Table in S5 Table). However, skin fibroblasts showed a very strong correlation of nCg enrichment per donor with that of both rCg and yCg motifs (S8B and S8C and S5d Table in S5 Table). This showed that the admixture of different mutational processes that cause C➔T base changes may influence the MEML count assigned to a specific process. To ensure that the observed differences in nCg, rCg, and yCg motif enrichments between different cell types were not driven solely by fluctuations in the background C➔T mutation load potentially contributed by multiple mutagenic processes, we decomposed enrichment per donor into the component mutation densities and visualized on log-log plot (S9 Fig and S5e Table in S5 Table, see Methods). We observed that the high enrichment per donor in chondrocyte clusters were driven by both relatively higher motif mutation densities and lower background mutation densities for nCg, rCg, and yCg motifs (S9 Fig). Skin fibroblast donors showed a broader spread with lower motif mutation densities and higher background mutation densities for all motifs (S9 Fig). We concluded that the differences in nCg, rCg, and yCg enrichment per donor observed between chondrocytes and skin fibroblasts are products of both mutation densities shifting independently and representative of underlying biological mechanisms rather than mathematical artefacts.

The other motifs detected in skin fibroblasts but not in chondrocytes were characteristic of UV mutagenesis. UV radiation forms covalent bonds between two adjacent pyrimidines to form CPDs and pyrimidine 6–4 pyrimidone (6–4PP) [44]. Resolution of cytosine CPDs by TLS polymerase results in C➔T mutations in the yCn context, while resolution of thymine dimers yields T➔C mutations in the nTt context [26,27]. We detected statistically significant enrichment of both UV motifs yCn➔yTn and nTt➔nCt in the skin fibroblast donors, but not in chondrocyte donors (Fig 4A and 4B and S5a Table in S5 Table). There was no statistically significant difference in the enrichment of yCn and nTt MEML between sun-protected and sun-exposed skin fibroblasts, however, sun-exposed skin fibroblasts showed significantly higher yCn and nTt MEML (Fig 4A and 4B and S5b Table in S5 Table). We also analyzed the sub-motif yCh which precludes the contributions of spontaneous deamination-mediated mutations in nCg motif from UV-associated yCn motif and detected its significant

enrichment only in skin fibroblast donors, with the sun-exposed skin fibroblasts showing a high burden of yCh MEML (Fig 4A and 4B and S5b Table in S5 Table). To investigate the significant enrichment of T > G mutations on the non-transcribed strand of only skin fibroblasts (Fig 1C), we further analyzed enrichment of mutations in yTn motif which encompasses CT and TT dimers. We detected multiple fibroblast samples with significant enrichment of yTn➔yGn motif (S5g Table in S5 Table). Spearman correlation analyses showed significant correlation of both UV motifs yCn➔yTn and nTt➔nCt with yTn➔yGn MEML, more prominently in sun-exposed skin fibroblast samples (S10A, S10B). These results indicate that the observed T > G strand bias in skin fibroblasts may be driven by the mutagenic bypass of persistent UV-mediated dipyrimi-dine lesions on thymines on the non-transcribed strand, likely through a REV1-mediated translesion synthesis mechanism that favors cytosine insertion opposite damaged thymines.

Statistically significant enrichment-adjusted MEML was also detected in skin fibroblasts for motifs that resemble deam-ination by cytidine deaminase enzymes (Fig 3). APOBEC leads to C➔T and C➔G mutations in the tCw context [43]. The enrichments of tCw➔tTw motif cannot be unambiguously assigned to APOBEC deamination, because of its unresolvable overlap with UV mutagenesis motif yCn (y = C or T). These overlapping trinucleotide motifs cannot be split into sub-motifs to remove the confounding effects of UV mutagenesis in skin fibroblasts, where we detected not only enrichment with yCn➔yTn motif of the UV mutagenic activity but also nTt➔nCt, minor UV mutagenic motif (Fig 4 and [26,27]). True APO-BEC mutagenesis is defined by enrichment of both tCw➔tTw and tCw➔tGw motifs. The latter is centered around C➔G base substitution, which excluded overlap with yCn➔yTn. However, APOBEC mutagenesis results not only in scattered mutations but also in C- or G- coordinated mutation clusters [43]. As described in [10], there was no or very few small C- or G-coordinated clusters in genomes of skin fibroblasts from the contribution of APOBEC mutagenesis in skin fibroblasts. Therefore, we could not conclusively detect APOBEC mutagenesis in skin fibroblasts.

Together, we conclude that motifs in both chondrocytes and skin fibroblasts can be associated with mutagenesis by two endogenous sources, spontaneous deamination of meCpG and exposure to small epoxides, and indicate higher activity of endogenous mutagenesis in chondrocytes. In contrast, motifs characteristic of mutagenesis by environmental UV expo-sure were detected only in skin fibroblasts.

## Motifs associated with endogenous mutagenic sources are clock-like in chondrocytes and skin fibroblasts

Next, we asked if the mutations associated with the identified mutagenic activities accumulate with age in chondrocyte and skin fibroblast donors. We performed Spearman correlation analyses between donor age and average MEML load for the motifs and sub-motifs shown in Fig 4 (S5f Table in S5 Table). We excluded sun-exposed skin fibroblasts from these analyses as the cohort only had two donors. Previous studies have reported age-dependent accumulation of both aTn and nCg (or nCg-like signature SBS1) associated mutation load in multiple cancers and non-cancer tissues, including skin fibroblasts [10,39,47]. We detected age-dependent accumulation of aTn motif MEML in chondrocyte single cells, but not in chondrocyte clones and sun-protected skin fibroblasts (Fig 5D and S5f Table in S5 Table). Analyses excluding donors with zero aTn MEML values showed significant age correlation for single chondrocyte and skin fibroblast donors in one-sided significance test (Fig 5D). Correlation of donor age with aTr MEML, sub-motif of aTn motif not confounded by the UV-associated nTt motif, also showed significant age correlation in skin fibroblasts (Fig 5D).

We also detected an age-dependent accumulation of nCg MEML in skin fibroblasts and chondrocyte single cells, but not in chondrocyte clones (Fig 5D and S5f Table in S5 Table). Notably, we did not detect any significant correlation between donor age and either rCg or yCg MEML for skin fibroblasts, despite their strong positive age correlation with nCg MEML (Fig 5D). Given rCg motif is purified from UV-mutagenesis, we would expect it to hold age correlation in tissues where nCg motif is clock-like. The significantly low rCg enrichment in skin fibroblasts, coupled with the lack of age correla-tion could be explained by a very low contribution of meCpG deamination in the C➔T mutations detected in skin fibroblast. In the same vein, we did not detect age-dependent accumulation of UV-associated motif MEMLs in skin fibroblasts owing to the episodic nature of UV mutagenesis (S5f Table in S5 Table).

While both clock-like motifs aTn and nCg show significant age correlation in the single chondrocyte cohort, such tendency was not detected in cohort with clonally expanded chondrocyte samples (S10C and S10D Fig). To assess if the lack of correlation in our dataset of chondrocyte clones could be a result of the restricted donor age range (55–86 years), we down-sampled the single-cell cohort (n = 17) to include only 10 donors falling within that same age window as the chondrocyte clone donors and evaluated correlation with donor age. We found that the significant age correlation previously detected in the full single-cell cohort was lost within this narrow age range (S10E Fig). This suggests that the lack of age correlation with aTn and nCg MEML in chondrocyte clones are likely attributable to the small sample size, narrow donor age distribution, or positive selection of clones that evaded damage-induced apoptosis.

Together, these analyses detected clock-like activity of exposure to small epoxides and $S_N2$ electrophiles, and spontaneous deamination of meCpG in both chondrocytes and skin fibroblasts.

## Single-base indels within long homonucleotide tracts dominate the chondrocyte indel spectra, whereas UV-induced deletions predominate in skin fibroblasts

Similar to the SNVs in chondrocyte clones, we did not detect a singular peak of indels between 45%-55% or ≥90% allele fraction (S11 Fig). As applied for SNVs, we only included indels that fell within this stringent filtering criteria to analyze true clonal indels in chondrocyte samples and compare with those in skin fibroblast samples, as well as with all indels detected in single cell chondrocyte samples. We detected a total of 525 indels in the genomes of 17 chondrocyte clones with sizes ranging from 1 bp to 24 bp (no clonal indel was detected in chondrocyte clone sample A06). Average indel load per donor was comparable in chondrocyte clones and sun-protected skin fibroblasts but higher in sun-exposed skin fibroblasts (Fig 6A and S6a and S6b Tables in S6 Table). The indel load was lower in single cell chondrocytes when compared to either chondrocyte or to skin fibroblast clones (Fig 6A and S6a and S6b Tables in S6 Table). While we observed a lower number of indels in single chondrocytes compared to chondrocyte clones (Fig 6A), we note that this difference may be influenced by the distinct variant-calling sensitivities and technical artefacts inherent to single-cell and clone sequencing. We also detected significant correlation between donor age and average indel load in both chondrocyte and skin fibroblast donors (Fig 6B and S6c Table in S6 Table). When compared across different cohorts within chondrocyte clones and single cells, we found no difference between healthy and OA chondrocyte clones (S12A Fig and S6b Table in S6 Table). Healthy chondrocyte single cell donors showed a higher indel burden with significant age-dependent accumulation as opposed to OA chondrocyte single cell donors (S12A and S12B Fig and S6b and S6c Tables in S6 Table).

To understand the patterns of indels in chondrocytes and skin fibroblasts, we generated indel matrices of 83 different indel types as utilized in the COSMIC small insertions and deletions (ID) signatures for seven sample groups using SigProfileMatrixGenerator [48]. We generated matrices of total indel calls from each of the seven sample cohorts and refitted the matrices with published indel signatures on COSMIC database [6], to detect their contributions in the indel profiles. Consistent with the previous observation in chondrocyte single cells [21], we detected large contributions of ID1 and ID2 signatures associated with replication slippage, as well as ID3 associated with DNA damage (including damage from tobacco smoking) in all chondrocyte sample groups (S13A Fig and S6d Table in S6 Table). ID8, signature associated with double-strand break repair by non-homologous end joining, was detected in small fractions across all chondrocyte cells as opposed to skin fibroblasts where it made up the largest fraction and was previously identified by agnostic signature extraction [26] (S13A Fig and S6d Table in S6 Table). ID15 with unknown etiology was also detected in all chondrocyte and skin fibroblast sample groups. Analysis of cosine similarity between indel matrices and COSMIC reference indel signatures detected strong cosine similarity (≥0.8) for ID1 in healthy single cell chondrocytes and ID8 in skin fibroblasts (S13B Fig and S6e Table in S6 Table). ID1, ID2, ID3, and ID15 are composed of 1 bp indels in homopolymer runs, while ID8 is constituted of deletions >5 bp. We performed further granular analyses in each of the seven cohorts, separating indels into 83 different indel types (S6f Table in S6 Table). Categorization into the same 83 indel types was previously used to identify endogenous and UV-associated indel types in skin fibroblasts [26].

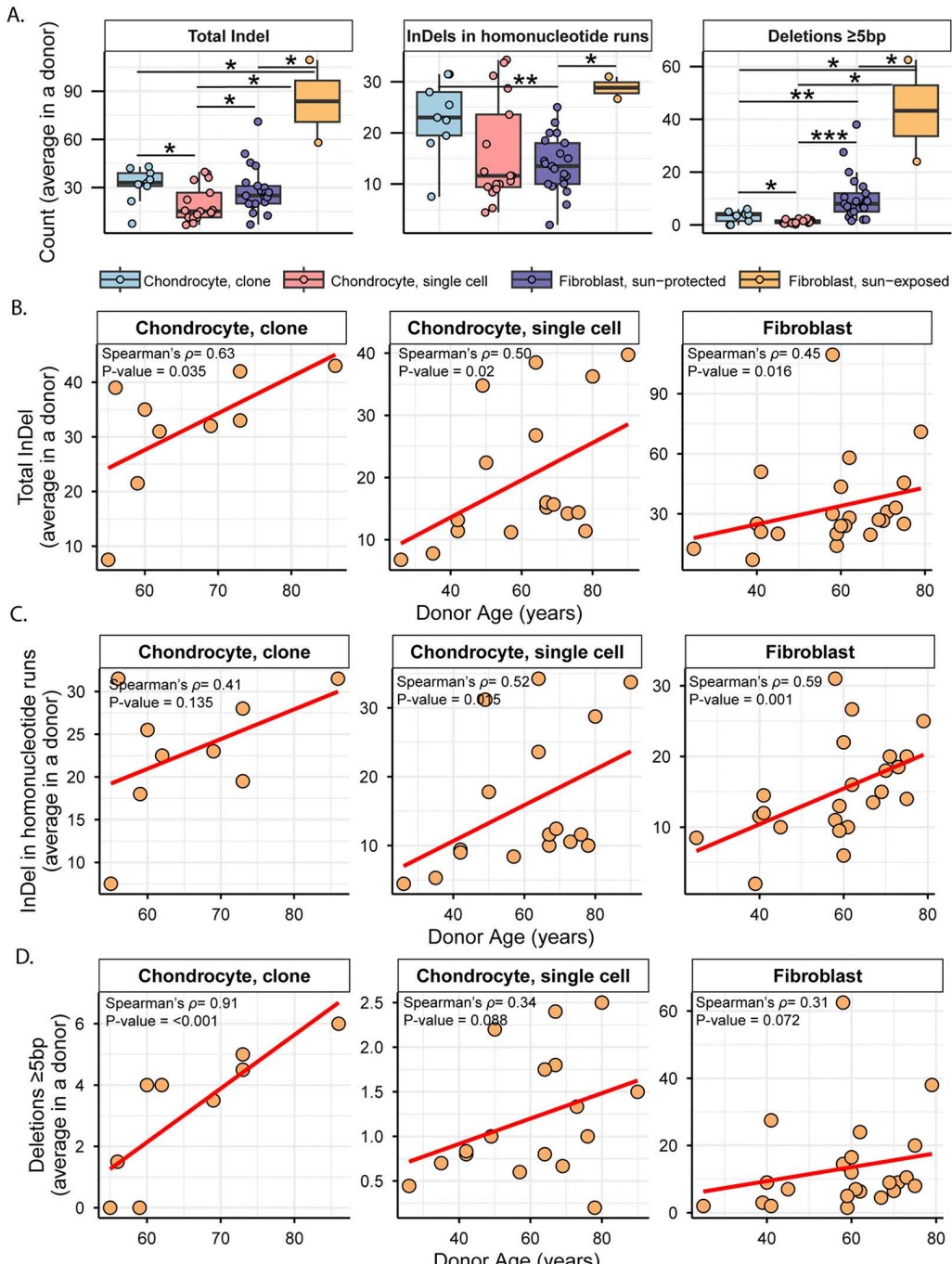

**Fig 6. Analyses of small indels in chondrocytes and skin fibroblasts. A.** Average count per donor was shown in a boxplots for total indels, indels in homonucleotide runs, and deletions ≥5 bp in different cell types. Each dot represents an individual donor. Asterisks represent statistical significance (Wilcoxon Rank Sum test) between connected cell types. Black solid connector lines indicate two-sided test, red dashed connector lines indicate one-sided test. *P value ≤ 0.05, **P value ≤ 0.01, ***P value ≤ 0.001. **B-D.** Average counts per donors plotted against donor age for total indels **(B)**, indels in homonucleotide runs **(C)**, and deletions ≥5 bp (D) within each cell and DNA group indicated. Correlation coefficient and one-sided p value from Spearman's correlation analyses are indicated on each plot. Red line indicates best-fit linear regression. Age correlations within sub-groups of cell types are shown in S12B Fig. Axis ranges for scatter plots in panels B-D were scaled to the specific range of data points within each cell type in order to maximize the resolution of internal trends and data distributions. For direct quantitative comparisons between groups using standardized scales, refer to the corresponding box plots in Fig 6A. All source data and p values from statistical analyses are available in S6 Table.

We found single base indels of thymine (and adenine in reverse complement) in homonucleotide runs consisting of ≥5 repeated single-nucleotide units to be the largest contributor in the chondrocyte clone indel matrix (deletions 17%, insertions 12% of all indels detected in chondrocyte clone samples) (S6f Table in S6 Table). Chondrocyte single cells also showed similar preference to insertion of thymine (and adenine in reverse complement) in homonucleotide runs consisting of ≥5 repeated single-nucleotide units as the highest contributor (25% of all indels detected in chondrocyte single cell samples). Unlike chondrocytes, single base deletions in dinucleotide homopolymers (10% in di-cytosine (and di-guanine in reverse complement), 7% in di-thymine (and di-adenine in reverse complement)) constituted the highest fractions of indels associated with homonucleotide runs in skin fibroblasts (S6f Table in S6 Table). The average load per donor for indels in homonucleotide runs were comparable between chondrocyte clones and single cells, while sun-protected skin fibroblasts showed a significantly lower load compared to chondrocyte clones (Fig 6A). Indels in homonucleotide runs showed a significant age-dependent accumulation in sun-protected skin fibroblasts and chondrocyte single cells, but not in chondrocyte clones (Fig 6C). Individual inspection of each chondrocyte cohort revealed age-dependent accumulation and a higher load of indels in homonucleotide runs in healthy chondrocyte single cells compared with their OA counterparts, while clonal chondrocytes showed no difference between healthy and OA (S12 Fig and S6b and S6c Tables in S6 Table).

Another class of indels which contributed the highest fraction in the indel matrices of skin fibroblasts and indicate the impact of UV-mutagenesis [26] was deletions of ≥5 bp not associated with microhomology (19%), followed by ≥5 bp deletions with 1 bp microhomology (16%) (S6f Table in S6 Table). Such indels were not overrepresented in the chondrocyte indel catalogues (3%-5% in chondrocyte clones, 2%-3% in chondrocyte single cells) (S6f Table in S6 Table). Skin fibroblasts showed a significantly high burden of deletions of ≥5 bp compared to that in chondrocytes (Fig 6A and S6b Table in S6 Table). Age-dependent accumulation of ≥5 bp deletions was not detected in skin fibroblasts or chondrocyte single cells, however, these deletions were seen to linearly accumulate with age in OA chondrocyte clones (Fig 6D and S12B Fig and S6c Table in S6 Table). Since the number of these indels were very low in chondrocytes, further investigation is required to identify true mechanisms. Together, these analyses revealed that chondrocytes endogenously accumulate higher loads of indels in long homonucleotide runs compared to skin fibroblasts, which accumulate indels in short homonucleotide runs and predominantly exhibit UV-induced large deletions.

## Functional annotation of somatic SNVs and InDels detected in chondrocytes

We annotated all somatic SNVs and indels detected in chondrocyte clones sequenced in this study for their functional effects using ANNOVAR [49]. We detected total 82 exonic SNVs and 4 exonic indels (S7a Table in S7 Table). We also detected 1 stop-gain and 56 nonsynonymous SNVs. We further annotated the SNVs and indels in both chondrocyte clones and single cells with Cancer Genome Interpreter [50]. We found one oncogenic variant in one chondrocyte clone, and 13 oncogenic variants across 10 chondrocyte single cells from 7 different donors (S7a Table in S7 Table).

To evaluate if the somatic variants in chondrocytes are enriched on OA-associated genes, we annotated all mutations in both clones and single cells for colocalization with 700 OA effector genes reported in a recent genome-wide association study (GWAS) (S7b Table in S7 Table) [51]. Previous study on chondrocyte single cells analyzed coincidence of mutations with 77 high confidence OA effector genes and found 135 mutations in 44 of the 77 genes [21]. Our analysis of both chondrocyte clones and single cells using a larger GWAS analysis identified 2,350 mutations in 469 OA effector genes (S7c Table in S7 Table).

## Structural variants in hotspots colocalize with common fragile sites in skin fibroblasts, but not in chondrocytes

We detected 73 structural variants in 18 chondrocyte clone samples in the form of deletions, duplications, insertions, inversions, and translocations (S8a Table in S8 Table) and compared that with numbers and features of structural variants previously found in skin fibroblast clones. The average number of rearrangements per donor ranged from 1 to 8 in chondrocyte clones and 1–11 in skin fibroblasts with sun-exposed skin fibroblasts showing significantly higher number

of structural variants compared to sun-protected skin fibroblasts (Fig 7A and S7b and S7c Tables in S8 Table). Deletions were the most predominant rearrangement detected in both chondrocytes and skin fibroblasts, followed by duplications (S8b Table in S8 Table). We detected 4 large insertions in OA chondrocyte clones while other cell types did not show insertions (S8a Table in S8 Table). Inversions were unique to skin fibroblasts except one inversion event detected in one chondrocyte OA clone. Translocations were detected in 1 chondrocyte OA clone, 3 sun-protected skin fibroblast samples, and 1 sun-exposed skin fibroblast sample (S8a Table in S8 Table). We did not detect any age-dependent accumulation of structural variants in chondrocytes, or in skin fibroblasts as was previously reported (Fig 7B).

We further annotated the structural variants for their incidence as hotspots, which were defined as breakpoint locations overlapping or within 1Mbp in different donors as previously established for sun-protected skin fibroblasts [26]. 6 of the 73

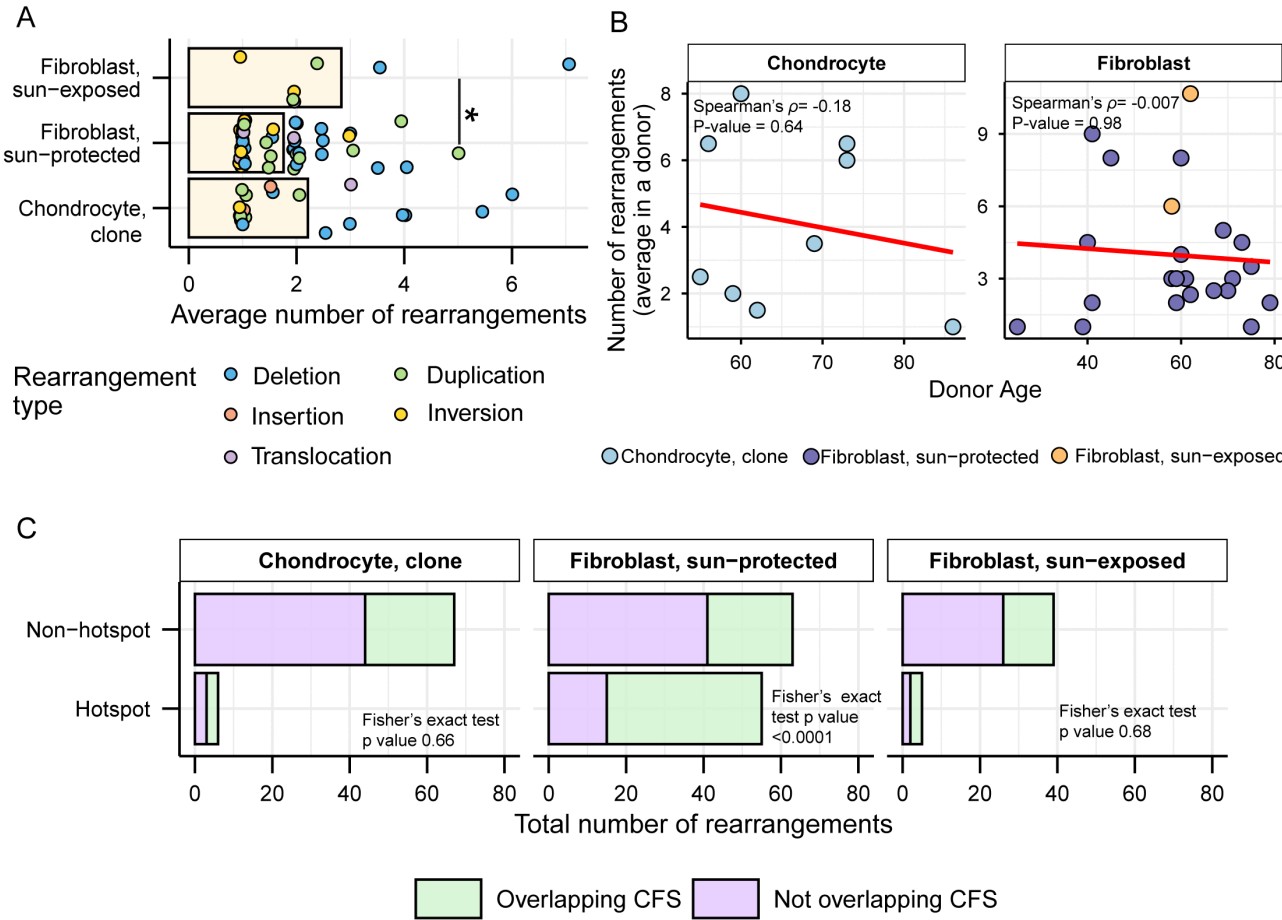

**Fig 7. Analyses of large rearrangements in chondrocytes and skin fibroblasts. A.** Mean number of rearrangements detected in different cell types. Each dot represents the average number of a rearrangement type indicated by color in individual donors. Asterisks represent statistical significance (Wilcoxon Rank Sum test) between connected cell types. Black solid connector lines indicate two-sided significance test. *P value ≤ 0.05. **B.** Average number of rearrangements detected in a donor plotted against donor age. Correlation coefficient and two-sided p value from Spearman's correlation analyses are indicated on each plot. Correlation analysis was not conducted for the sun-exposed donors but was still displayed in the plot, as there were only two donors in that group. Axis ranges for scatter plots in panel B were scaled to the specific range of data points within each cell type in order to maximize the resolution of internal trends and data distributions. For direct quantitative comparisons between groups using standardized scales, refer to the corresponding bar plot in panel **A. C.** Total number of rearrangements detected in different biospecimens stratified by whether they overlap with rearrangement hotspots and human common fragile sites (CFS). P values from Fisher's exact test are indicated on each panel. All source data and p values from statistical analyses are available in S8 Table.

structural variants detected in chondrocyte clones were colocalized in hotspot regions (S8d Table in S8 Table). We also analyzed if the structural variants detected in chondrocyte clones intersect with human common fragile sites (CFS) reported in the HumCFS database [52]. We found 26 of the 73 rearrangements detected in chondrocyte clones to coincide with known CFS (S8d Table in S8 Table). Our previous study in sun-protected skin fibroblasts had identified statistically significant coincidence of rearrangements in hotspots with CFS [26]. We did not detect such a tendency in the chondrocyte genomes (Fig 7C). Since CFS are identified as chromosome positions vulnerable to breakage in response to inhibited DNA strand elongation and possibly to other kinds of replication stress [53], association between breakpoints and CFS locations may be indicative of skin fibroblast chromosomes being more prone to breakage during DNA replication as opposed to hypo-replicative chondrocytes.

## Discussion

In this study, we sequenced and analyzed 18 clonally expanded chondrocyte samples from donors with and without OA. We validated the identified burden and sources of somatic mutations in chondrocyte clones by analyzing 100 single chondrocyte cells sequenced in a recent study [21]. To characterize the somatic mutagenesis landscape of differentiated mesenchymal cell types, we compared the genomic instability features of chondrocytes with those of clonally expanded sun-exposed and sun-protected skin fibroblasts sequenced in our previous studies [26,27]. Our work provides a baseline of different somatic genome instability features detectable in chondrocytes and speculates on their associated mechanisms. The following discussion addresses major features of chondrocyte genome change accumulation and compares these features with those in skin fibroblasts (summarized in Table 1), which share a mesodermal origin but diverge early in development to inhabit vastly different organism niches.

Analyses of mutation catalogues using both mutational signature refitting and motif-centered analyses revealed only endogenously accumulated mutations in chondrocytes, as opposed to skin fibroblasts which also showed a high burden of UV-induced mutations (Figs 2 and 3 and Table 1). This distinction is consistent with the anatomical location and physiological functions of the two cell types. Signature refitting with chondrocyte mutation catalogues in this study, as well as in a previous study on single chondrocyte genomes combined with *de novo* signature extraction [21], revealed high contributions of clock-like signature SBS5 whose etiology is yet unknown (Fig 2 and Table 1). Importantly, SBS5 includes peaks for all 96 possible trinucleotide mutational motifs and therefore likely contains many if not all of the known mutagenic mechanisms that are analyzed in this study (S5 Fig). The use of knowledge-based motif-centered analyses in parallel with signature fitting enabled the detection of two distinct mutational processes, deamination of meCpG and exposure to small epoxides and $S_N2$ reacting electrophiles, operating in chondrocyte genomes as well as in skin fibroblasts (Fig 3 and Table 1). Both of these mutagenic processes are well documented in many cancers and normal tissues using motif-centered and/or mutation signature analyses [6,10,39]. This study provides the first evidence of the activities of these two mutational processes in chondrocytes, proving utility of motif-centered analyses in uncovering biologically interpretable mutational mechanisms that may not be resolved by conventional signature analyses in tissues with low mutation burdens. Analyses of sub-motifs also allowed the separation of mutagenic activities in skin fibroblasts that cause the same base substitutions in overlapping nucleotide contexts (Figs 4, 5, and S8 Fig).

Notably, we found higher activities of the two endogenous mutational processes in chondrocytes compared to that in skin fibroblasts, as evidenced by the higher enrichment of the associated mutational motifs and sub-motifs (Fig 4 and Table 1). aTn and its sub-motif aTr, both of which showed modest but significantly increased enrichments in chondrocyte clone and single cell donors compared to skin fibroblast donors, are associated with exposure to small epoxies and possibly a broader class of $S_N2$ electrophiles [39]. Reactive small epoxides are systemically released as byproducts of xenobiotic metabolism primarily performed in liver by CYP epoxygenases [54]. Both chondrocytes and skin fibroblasts also produce epoxides called Epoxyeicosatrienoic Acids (EETs), which have been shown to form DNA adducts *in vitro*, from CYP450-mediated metabolism of arachidonic acid [55–57]. Alongside the endogenous exposure in skin fibroblasts, epoxides and $S_N2$ electrophiles can also be introduced by dermal absorption of industrial chemicals [58]. Analyses of

**Table 1. Summary of comparisons between chondrocytes and skin fibroblasts.**

| Mutation class | Feature | Chondrocytes | Skin fibroblasts |
|---|---|---|---|
| SNV | Average load in a donor | - Clock-like<br>- Low burden [1] | - Not clock-like<br>- High burden[2] |
| | accumulation per cell division | - Lower rate[1] | - Higher rate[2] |
| SBS (SNV) mutational signature | Signature refitting | - High contribution of SBS5<br>- No SBS7a, SBS7b<br>- SBS1[3], No SBS16 | - No SBS5<br>- High SBS7a, SBS7b,<br>- No SBS1; No SBS16 |
| Mutational motif | Small epoxide exposure (aTn) | - Ubiquitous presence<br>- Higher enrichment[1]<br>- Clock-like[4] | - Ubiquitous presence<br>- Lower enrichment[2]<br>- Clock-like |
| | Spontaneous meCpG deamination (nCg) | - Ubiquitous presence<br>- Higher enrichment[1]<br>- Clock-like[4] | - Ubiquitous presence<br>- Lower enrichment[2]<br>- Clock-like |
| | UV exposure (yCn, nTt) | - Not detected | - Ubiquitous presence |
| InDel | Average load in a donor | - Clock-like | - Clock-like |
| | in homonucleotide runs | - Higher fraction within indels[1]<br>- In long runs (≥5 bp)<br>- Clock-like[4] | - Lower fraction within indels[2]<br>- In short runs (1 bp)<br>- Clock-like |
| | Deletions ≥5 bp; w/o repeats | - Lower fraction[1]<br>- Clock-like[3] | - Higher fraction[2]<br>- Not clock-like |
| Structural variants | Average load in a donor | - Not clock-like | - Not clock-like |
| | In hotspot | - Lower fraction[1]<br>- Low overlap with CFS | - Higher fraction[2]<br>- High overlap CFS |

1.Compared to skin fibroblast cohorts.

2.Compared to chondrocyte cohorts.

3.Only in chondrocyte clones.

4.Only in chondrocyte single cells.

yeast ssDNA system and mouse embryonic fibroblasts (MEFs) exposed to small epoxide glycidamide showed preference for A➜G mutations in nAt motif (reported in conventional pyrimidine language as T➜C mutations in aTn motif) [39]. nAt➜nGt mutations were found significantly enriched on the non-transcribed strands of MEFs and majority of the PCAWG cancer genomes [39]. Consistent with these findings, we detected significant enrichment of T➜C mutations on the transcribed strands (A➜G mutations on the non-transcribed strands) of both chondrocyte and skin fibroblast cell types (Fig 1C), likely suggesting a similar mechanism that causes more mutagenic lesions to adenines by ssDNA-specific damaging agent in ssDNA of the non-transcribed strand formed in R-loops (See analysis supporting this suggested mechanism for A➜G mutagenesis by small epoxides in [39]). Alternatively, mutagenesis biased towards non-transcribed strand can stem from transcription-coupled nucleotide excision repair (TC-NER) preventing mutagenesis on transcribed strand [59]. Transcription coupled repair can also be invoked to explain the strong bias towards non-transcribed strand displayed by C➜T mutations in skin fibroblasts but not in chondrocytes. C➜T mutations are the most prominent mutation class caused by UVB and UVC, and the role of transcription-coupled repair is well established for resolution of UV-induced bulky CPD lesions [60]. C➜T transcription bias specific to skin fibroblasts is in good agreement with the presence of UV-associated mutational motifs and indels >5 bp only in skin fibroblasts but not in chondrocytes that are well hidden from UV-irradiation inside the body (Figs 4 and 6 and Table 1)

We also detected a significant and multi-fold higher enrichment of nCg mutational motif associated with deamination of meCpG in chondrocytes compared to skin fibroblasts (Fig 4). C➜T mutations in nCg motif overlap with that in yCn motif associated with UV radiation-a strong mutator in skin fibroblasts. Analyses of sub-motifs of both nCg and yCn motifs

detected both meCpG deamination and UV-induced mutations in skin fibroblasts. In contrast, chondrocytes only exhibit meCpG deamination. This divergence is further substantiated by the distinct strand asymmetry of C➔T mutations, which are slightly enriched on the transcribed strand of healthy and non-lesional OA chondrocyte sample groups and consistently enriched on the non-transcribed strand across all skin fibroblast sample groups (Fig 1C). These patterns align with the activities of known DNA repair pathways, as T:G mismatches from spontaneous meCpG deamination are primarily resolved by base excision repair (BER), whereas UV-induced bulky CPDs are primarily removed by TC-NER [61,62]. BER pathway is known to be coordinated with DNA replication and cell cycle progression [63] and thus may have slow DNA repair turnover on non-cycling mature chondrocytes, resulting in persistent accumulation of T:G mismatches and eventual fixation of nCg➔nTg mutations. Spontaneous deamination of meCpGs is further exacerbated by reactive oxygen and nitrogen species (RONs) [61,64]. Chondrocytes produce RONs as part of normal signaling and homeostasis, and an imbalance in RON production and anti-oxidant capacity of chondrocytes have been implicated with oxidative stress-induced chondrocyte death and cartilage degradation [65,66]. Because chondrocytes have low turnover and are largely hypo-replicative, sustained exposure to RONs may promote accumulation of unrepaired meCpG deamination over time. While skin fibroblasts also produce RONs under normal physiological conditions and in response to UV radiation [67], their higher proliferative capacity and continued access to DNA repair machineries may limit the persistence of deamination lesions. Together, these indicate a plausible mechanistic basis for the higher enrichment of endogenous mutational processes observed in chondrocytes compared to skin fibroblasts.

Indel spectrum in chondrocytes was heavily leaning towards single bp deletions in large homonucleotide runs (Fig 6 and Table 1). Indels in homonucleotide runs (termed as microsatellites in eukaryotic genomes) are prone to spontaneous replication slippage rather than to DNA lesion induced events [68]. While bulges resulting from DNA polymerase slippage are efficiently repaired by mismatch repair or by DNA polymerase proofreading, longer runs escape proofreading and thus are more prone to slippage-generated small indels within a repeat stretch [69–72]. Higher fraction of such indels in homonucleotide runs in chondrocytes vs skin fibroblasts is in agreement with higher impact of endogenous SNV mutagenesis (nCg and aTn) in chondrocytes as compared with skin fibroblasts. Another type of indels observed in chondrocytes as well as in skin fibroblasts was deletions >5 bp not associated with repeats. These deletions prevailed in skin fibroblasts and constituted a much lower fraction in chondrocytes. The likely cause of these deletions are double-strand breaks (DSB) that, in principle, can be induced by environmental or by endogenous DNA damage. Such breaks can be accurately repaired by homologous recombination or result in deletions, if repair occurs by one of the end-joining mechanisms [73,74]. If a bulky lesion induced by UV escapes nucleotide excision repair, it could impede replication and lead to a DSB. Consistent with the exclusive presence of UV-mutagenesis base substitution motifs, deletions >5 bp were more prominent in skin fibroblasts than in chondrocytes (Fig 6 and Table 1)

We note that unlike skin fibroblast donors, chondrocyte donors showed an age-dependent accumulation of SNVs (Fig 1B). This is consistent with our findings from motif-centered analyses, as chondrocytes only accumulate mutations from endogenous sources over time. In support of this notion, we also detect age-dependent accumulation of aTn and nCg motif MEML, total indels, and homonucleotide indels in single chondrocyte donors. Motif-age correlations were lacking in chondrocyte clone donors, likely attributable to the small number of donors with narrow age distribution, or the possible selective expansion of clones with reduced accumulation of these mutational motifs with aging. Contrary to chondrocytes, skin fibroblasts are additionally exposed to episodic environmental mutagens that can obscure age-dependent genome changes. Indeed, we detect significant age correlated accumulation of nCg MEML, aTr MEML, and indels in homonucleotide runs, all predominantly arising from endogenous mutational processes, in skin fibroblast donors. However, we do not see age-dependent accumulation of yCn MEML and deletion larger than 5 bp which are characteristic of UV-induced mutations. In addition to the unique individual exposures during chronological aging, genotypes of these chondrocyte and skin fibroblast donors may also dictate the rate of accumulation of both endogenous and environmentally induced mutations.

The analyses performed in this study for chondrocyte clones and single cells showed a similar total load of SNVs, indels, large rearrangements, and mutational motifs between OA and non-OA chondrocytes. The prior analysis of single chondrocytes showed a similar burden when comparing OA lesion and OA non-lesion cells from four donor samples that had both, but a significant decrease in burden when comparing OA lesion chondrocytes non-OA (necrosis of the femoral head) cells [21]. Together, these two sets of data and analyses argue against an increased mutational burden in OA chondrocytes. One explanation is that there was selective survival and/or colony expansion of OA chondrocytes with more modest burden as compared to other chondrocytes from the same tissue that were not captured in sequencing. This interpretation of survival bias is further supported by the telomere analyses, which showed capturing of progenitor cells with longer telomeres from older donors regardless of OA status. It is also possible that some selection bias happens *in situ*, as apoptosis may eliminate damaged cells whereas the cluster formation that occurs in later stages of OA may allow for preferential division of cells with a lower number of mutations [75].

The overall burden and types of mutations in chondrocytes are consistent with studies from other cell types that are protected from high levels of external mutagens, including hypo-replicative cells such as cardiomyocytes and neurons. The finding of increased mutagenesis with age in these cell types suggests that mutagenic burden may be driven more by endogenous processes within the cell as compared to division events [76]. This has implications for the potential role of DNA damage and mutagenesis in OA. While DNA damage is recognized for playing a central role in aging [77], the contribution of somatic mutagenesis in diseases other than cancer is only beginning to be recognized [4,78], probing the possibility that dysfunctions caused by particular mutations could play a role in OA. Furthermore, the mutagenic landscape may serve as a "canary in the coal mine" that shows the extent of DNA damage in long-lived cells such as chondrocytes. While most of the damage may be repaired before being fixed as a mutation, quantitative assessments of mutational burden may give important information on the risk factors for age-related OA and support therapeutic approaches to boost DNA repair capacity or mitigate senescence that may result from excessive damage that overwhelms the repair capacity in joint tissues [79,80].

Together, our study provides a comprehensive record of genome instability features in two mesenchymal cell types and identifies yet unreported specific mutagenic mechanisms operating in chondrocyte genomes. This knowledge about sources of somatic mutagenesis can lead to therapeutic targets aimed at resolving mutagenic lesions or increasing repair in OA chondrocytes.

## Materials and methods

### Ethics statement

The tissue samples included in this study were de-identified and appropriate use has been determined by the University of North Carolina Institutional Review Board. Cartilage tissue was sourced either from cadaveric donors through the Gift of Hope Organ & Tissue Network and Rush University (protocol number: ORA 08082803IRB01AM02) or from waste tissue obtained from surgeries performed by the Department of Orthopaedics within the University of North Carolina School of Medicine (study number 14–0189). The University of North Carolina Institutional Review Board reviewed our use of human tissue and determined it did not meet the criteria for human subjects research and so formal consent was not required.

### Sample collection and processing

Unless otherwise noted, isolation of human chondrocytes from articular cartilage and clonal expansion was performed as in our previously published protocols [33,81]. Cartilage from OA donors was taken from areas that were considered "non-lesional" in Ren *et al.* [21]. Donor cartilage from cadaveric source was selected from individuals without prior known diagnosis of osteoarthritis and only minimal macroscopic damage (scores between 0–2 in modified Collins grade) [82].

Demographic information of the donors is provided in S1 Table. Briefly, human cartilage tissue was cut into small pieces using surgical-grade blades and were subjected to digestion in 2 mg/ml Pronase (CAS # 9036-06-0, Millipore Sigma) in the absence of serum for 1h. Digestion was completed using 0.36 mg/ml Collagenase P (Roche) in 5% serum for 18-20h. Isolated chondrocytes were cultured at a density of 100,000 cells/cm². Following a 3–4 day recovery in monolayer culture, cells were harvested with TrypLE Express and plated at a density of approximately 10 cells/cm² in 60-mm cell culture treated dishes. Low-density cultures were maintained at low oxygen tension (3% $O_2$) for faster growth while minimizing the number of culture-induced mutations. Clones were harvested at 10–20 days by pipette using phase-contrast microscopy guidance and then passaged several times until more than 1 million cells were present.

Genomic DNA from approximately 1 million cells were harvested to generate genomic DNA for library construction using Qiagen DNAeasy Blood and Tissue kit (69504). In addition to expanded clones, genomic DNA isolation was performed from the bulk population of cells of each donor. Column-purified genomic DNA was quantified using Qubit HS dsDNA kit (Invitrogen, Q32851) and was submitted for sequencing library preparation.

## Whole genome sequencing and detection of somatic variants in sequenced chondrocyte clones

Libraries were prepared for 27 chondrocyte samples (18 clones and 9 bulks) using Watchmaker DNA Library Prep Kit with Fragmentation (Watchmaker Genomics). 12 chondrocyte samples were sequenced on NovaSeq 6000 platform (Illumina) and 15 were sequenced on DNBSEQ-T7 platform (Complete Genomics). All samples were sequenced at 34x to 114x average coverage with mean insert size of 144 bp to 322 bp. FASTQ files of clone and bulk samples were aligned to the hg19 genome using BWA-MEM [83]. Two mutation callers, VarScan2 [84] and Strelka2 [85], were used to detect clone-specific mutations using bulk chondrocytes as germline surrogates. Mutations that passed as somatic were filtered for 3x total coverage and ≥10 reads supporting the SNV position. Consensus mutation calls from both callers were further filtered to remove mutations that were present in dbSNP138 database and that overlapped with SimpleRepeats track in the UCSC Genome Browser. Final mutation calls were filtered based on allele fractions of 45%-55% or ≥90% to retain true clonal mutations. From the mutation catalogue of 18 chondrocyte clones, we detected 82 mutation pairs that were repeated in the individual mutation catalogues of at least two chondrocyte samples. 68 of these repeat mutation pairs were shared between two samples derived from the same donor and contributed to 0.6%-3% of all detected in those donors. 14 of the repeat mutation pairs were shared between samples from different donors. Since identical calls were only a small fraction and each pair of identical calls fell into the same range of allele fractions (45–55% or 90–100%) they were likely to represent infrequent incidence of identical mutations. We retained these identical mutations in the individual chondrocyte mutation catalogues. Indels were called using the same callers and filtering criteria as applied for SNVs except an additional filter to remove indels that overlap with RepeatMasker track in the UCSC Genome Browser. We also detected 3 repeat indel pairs with allele fractions between 45%-55%. Two of the repeat indel pairs were shared between two samples of two donors, while one pair was shared between samples of different donors. Indels in the sun-exposed fibroblasts were called using SvABA [86] and filtered as previously described for sun-protected fibroblasts [26]. BAM files for all bulk and clone chondrocyte samples, and MAF files containing SNVs and indels filtered for allele fractions of 45–55% or 90–100% are deposited in dbGaP study phs001182.v3. Because data were generated using both NovaSeq and DNBSEQ platforms, we assessed potential batch effects by comparing key metrics between the two platforms using two-tailed Mann-Whitney U tests. We found no significant differences in average sequencing coverage (p value 0.1011), total SBS burden (p value 0.5148), or the enrichment-adjusted MEML of aTn and nCg motifs (p values 0.4457 and 0.9483 respectively).

Delly (v 1.3.2) [87] was used to identify structural variants in the form of deletions, duplications, insertions, inversions, and translocations in chondrocyte clones Somatic calls with 'IMPRECISE' or 'LowQual' tags were removed. Novel junctions were further filtered by breakpoint allele fractions. Deletions, inversions, insertions, and translocations were considered clonal if the novel junctions had 40%-60% or ≥90% of supporting reads and no reads supporting the junctions in

the corresponding bulk genomes. Duplications were considered clonal if 20%-60% reads supported the novel junctions in clone with no reads supporting the junctions in corresponding bulk genomes. Structural variants within 1Mb distance of each other or with the same genomic coordinates in different donors were annotated for localization in 'hotspots'. Structural variants were further annotated for overlap with common fragile sites available from the HumanCFS database [52]. SNV and Indel calls for single cell chondrocyte samples were used as provided in [21]; SNV, InDel, and SV calls in fibroblast clones were taken from [26,27].

## Estimation of rate of SNV accumulation per genome and per cell generation

Data for rate of mutation accumulation in fibroblasts were collected from previous study [27]. Rate of mutation accumulation in chondrocyte clones sequenced in this study, and additional sun-protected fibroblasts reported in [26] were analyzed as previously described [27]. The number of cell generations was calculated based on the estimated telomere loss in chondrocyte samples since newborn. The average length of the terminal restriction fragments (TRFs) has been estimated to be ~11kb in newborns [88]. We estimated the average telomere sizes of the chondrocyte samples using TelSeq [89]. 1kb was added to the estimated telomere size to account for the retention of non-telomeric DNA between the restriction site and telomeric repeats in TRF analysis. Since the chondrocytes samples were expanded in culture for approximately 20 generations, we further added 2kb to the estimated telomere length to calculate the telomere length of the progenitor cell that gave rise to the clone. The length of telomere lost was calculated by subtracting the telomere length of the progenitor cells from that of newborns, and the number of cell divisions since birth was estimated to be 33–74. Mutation rate per genome per cell division was calculated using the following formula:

$$Rate_{per\ genome\ per\ cell\ division} = \frac{Total\ mutation\ count}{Number\ of\ cell\ divisions}$$

We also calculated the mutation rate per nucleotide per cell division by dividing $Rate_{per\ genome\ per\ cell\ division}$ with diploid human genome in nucleotides ($6 \times 10^9$ nt, S2c Table in S2 Table) Mutation per genome per year was calculated by dividing the average donor mutation load with donor age (S2c Table in S2 Table).

## Detection of transcriptional bias of SNV accumulation

Transcriptional strand bias of SNV accumulation was calculated using the MutationalPatterns R package [90]. mut_strand() function was used to annotate strand information and mut_matrix_stranded() function was used to generate 192 mutation matrix. strand_occurrences() function was used to calculate the total number of mutations and relative contribution corresponding to each base type, strand, and biospecimen type. strand_bias_test() function was used to perform two-sided Poisson test and detect significant stand asymmetry. plot_strand() was used to generate relative contribution plot and plot_strand_bias() function was used to plot log2 ratio of mutations on transcribed and non-transcribed strand.

## Analysis of COSMIC signatures

SNV and indel data were pooled for seven sample groups to generate cell type-specific SBS96 profiles (for SNVs) and ID83 profiles (for indels). MutationalPatterns R package [90] was used to calculate contribution of published COSMIC signatures and their cosine similarities with the variant profiles of different cell types. Signature refitting was performed to calculate contribution of published COSMIC SBS and ID signatures in the SBS and ID catalogues of all cell types using fit_to_signatures() function. Signatures contributing >5% of the mutations in a cell type were selected as the new signature set for refitting the mutation catalogue of that cell type. Cosine similarities of the SBS and ID profiles respectively with known SBS and ID signatures were identified using the cos_sim_matrix() function.

## Analysis of mutational motifs

A list of knowledge-based, experimentally validated mutational motifs was curated from literature and in-house experiments (S4 Table). Enrichment of these mutational motifs in the mutation catalogues of individual donors, as well as enrichment and minimum estimate of mutation load (MEML) attributable to the associated mutagenic process in individual chondrocyte clone, chondrocyte single cell, and fibroblast samples were calculated as previously described [10,26,27,39,43]. Enrichment is calculated as the ratio of two ratios: the ratio of mutations in motif to all mutations, and the ratio of the number of motifs to all bases in genomic context of ±20 bases surrounding mutations ("Genome" in the below formula). For this calculation, context is defined as the +/- 20 bases surrounding the mutated base. The use of genomic context immediately surrounding the mutated nucleotide rather than the whole genome sequence helped to concentrate on the sections of the genome that were actually sequenced and on localized preference of mutagenesis stemming from small-range preference of lesion occurrence, and to alleviate confounding effects of other factors, including epigenomic features influencing overall mutagenesis within greater genomic regions. This is shown in a simplified formula below.

$$Enrichment = \left( \frac{Mutations\ in\ Motif}{All\ Mutations} \right) : \left( \frac{Motifs\ in\ Genome}{Bases\ in\ Genome} \right)$$

Enrichment of specific mutational motifs in the mutation catalogues of individual donors and samples was calculated by the following formula (using nCg motif as an example).

$$Enrichment(nCg \rightarrow nTg) = \frac{[Mutations_{nCg \rightarrow nTg}] \times [Context_c]}{[Mutations_{C \rightarrow T}] \times [Context_{ncg}]}$$

where capital C is the mutated base in nCg motif. Reverse compliments were included in calculations, and mutation ≤10 bp apart from each other were excluded as 'complex' mutations potentially rising from the activities of translesion polymerases. Context in this formula is defined as ±20 bases flanking the mutation. Motif mutation density was calculated as

$$Motif\ mutation\ density = \frac{Mutations_{nCg \rightarrow nTg}}{Context_{ncg}}$$

Background mutation density was calculated as

$$Background\ mutation\ density = \frac{Mutations_{C \rightarrow T}}{Context_c}$$

To calculate if the enrichment is statistically significant, a Fisher's exact test was performed by comparing the ratio of mutations in a motif ($Mutations_{nCg \rightarrow nTg}$) and mutations that do not conform to the motif ($Mutations_{C \rightarrow T} - Mutations_{nCg \rightarrow nTg}$), versus the ratio of unmutated bases in the context that are either in the trinucleotide motif ($Context_{ncg}$) or not ($Context_c - Context_{ncg}$). For individual samples with statistically significant enrichment of a mutational motif after correction for multiple hypothesis using the Benjamini-Hochberg method (Enrichment >1, q ≤ 0.05), we calculated the minimum number of mutations that can be attributable to the mutagenic process associated with the mutation motif using the following formula:

$$MEML(nCg \rightarrow nTg) = \frac{[Mutations_{nCg \rightarrow nTg}] \times [Enrichment_{nCg \rightarrow nTg} - 1]}{[Enrichment_{nCg \rightarrow nTg}]}$$

A MEML value of zero is assigned for samples with no detected enrichment (enrichment <1) or with statistically insignificant enrichment (enrichment >1, q > 0.05) of a motif.

### Annotation of SNVs and indels

We used Annovar [49] to functionally annotate all clonal SNVs and indels in chondrocytes. refGene track from UCSC Genome Browser was used to annotate changes in protein sequences and their functional consequences. All protein-altering variants were annotated as cancer driver or passenger mutations using the Cancer Genome Interpreter [50]. All variants were further annotated for their localization on OA effector gene as identified by a recent GWAS meta-analysis [51].

### Statistical analyses

All statistical analyses were performed in R statistical software (version 4.4.0) via RStudio except for the Fisher's exact test in Fig 7c, which was performed in GraphPad Prism (version 10.0.0). One-sided tests were used to evaluate p values for correlation of total mutation burdens with age, as mutations are expected to increase with age. To further quantify the relationship between donor age and mutation burden in chondrocytes, we fitted linear mixed-effects (LME) models using the lme4 R package [91]. Separate models were constructed for chondrocyte clones and single cells. The response variable in both models was total substitutions per sample as reported in S2a Table in S2 Table. Age was included as a fixed effect and Donor ID was included as a random intercept in the models as "Total substitution ~ Age + (1 | Donor)".

### Data visualization

All data visualizations were performed using ggplot2 R package [92] and Adobe Illustrator [93]. Elements licensed as 'Public Domain' from NIH BioArt Source (https://bioart.niaid.nih.gov/) were used to generate S1A Fig (NIAID Visual & Medical Arts 143, 152, 153, 154, 244, 303, 404).

### Supporting information

**S1 Fig. Schema of chondrocyte colony isolation and DNA sequencing performed in this study. A.** Bulk chondrocytes were isolated from donor cartilage tissues and seeded at low density to generate single-cell derived colonies. Colonies were propagated for less than 20 generations to isolate and sequence DNA. **B.** Consensus somatic variants (SNVs and InDels) from two callers were generated using bulk chondrocytes as the matched normal. Somatic calls were further filtered to retain high confidence clonal variants. Elements from NIH BioArt Source (https://bioart.niaid.nih.gov/) licensed as "Public Domain" were used to generate panel A.
(TIF)

**S2 Fig. Distribution of allele fraction of SNVs detected in chondrocyte samples sequenced in this study.** Allele fractions of all SNVs before filtering are plotted in bins of five. Samples originating from the same donor are indicated by color.
(TIF)

**S3 Fig. SNV load and accumulation rate per cell division in individual genomes. A.** Total SNV load within each cell type is shown in boxplot. Each dot represents an individual sample. Asterisks represent statistical significance (Wilcoxon Rank Sum test) between connected cell types. Black solid connector lines indicate two-sided test, red dashed connector lines indicate one-sided test. *P value ≤ 0.05, **P value ≤ 0.01, ***P value ≤ 0.001. **B.** Donor mean mutation load plotted against donor age for each sub-group of cell type indicated. Correlation coefficient and one-sided p value from Spearman's correlation analyses are indicated on each plot. Red line indicates best-fit linear regression. Axis ranges for scatter

plots were scaled to the specific range of data points within each cell type in order to maximize the resolution of internal trends and data distributions. For direct quantitative comparisons between groups using standardized scales, refer to the corresponding box plots in S3A Fig. **C.** Mutation accumulation rate per cell division in different cell types shown in boxplot. Each dot represents an individual sample. Asterisks represent statistical significance (Wilcoxon Rank Sum test) between connected cell types. Black solid connector lines indicate two-sided test; red dashed connector lines indicate one-sided test. *P value ≤ 0.05, **P value ≤ 0.01, ***P value ≤ 0.001. All source data and p values from statistical analyses are available in S2 Table. **D.** Residual values per sample from LME model representing portion of total mutations that cannot be explained by age. A positive residual indicates a higher mutation load, and a negative residual indicates a lower mutation load than expected for that specific age. P-values for comparisons between healthy and OA were calculated using two-sided Wilcoxon rank-sum tests. **E.** Donor-average estimated telomere length plotted against donor age for each sub-group of cell type indicated. Correlation coefficient and two-sided p value from Spearman's correlation analyses are indicated on each plot. Red line indicates best-fit linear regression. Axis ranges for scatter plots were scaled to the specific range of data points within each cell type in order to maximize the resolution of internal trends and data distributions. **F.** Donor-average estimated number of cell divisions plotted against donor age for each sub-group of cell type indicated. Correlation coefficient and two-sided p value from Spearman's correlation analyses are indicated on each plot. Red line indicates best-fit linear regression. Axis ranges for scatter plots were scaled to the specific range of data points within each cell type in order to maximize the resolution of internal trends and data distributions.
(TIF)

**S4 Fig. SBS96 spectrum of all cell types analyzed in this study.** Mutations were pooled for all samples within a cell type and sorted in 96 trinucleotide channels to generate SBS96 spectrum profiles.
(TIF)

**S5 Fig. Mutational profiles of COSMIC reference SBS signatures.** Plots are shown for SBS5 signature with peaks across all 96-trinucleotide spectrum, as well as signatures with distinct peaks that resemble knowledge-derived trinucleotide mutational motifs.
(TIF)

**S6 Fig. Mutational motif prevalence and enrichment. A.** Percentage of donors with zero and non-zero MEML of indicated knowledge-based motifs in different allele fractions of SNVs detected in chondrocyte clones sequenced in this study. **B.** Fold enrichment of mutational motifs per donor for motifs indicated on panel labels are shown for different sub-groups of chondrocyte cell types. The blue dashed line indicates level of no enrichment. Each dot represents an individual donor. Wilcoxon Rank Sum test was performed across different chondrocyte sub-groups within each DNA group, and no statistically significant difference was found. All source data and p values from statistical analyses are available in S5a Table in S5 Table.
(TIF)

**S7 Fig. aTn and nCg contributions in AT and CG-pair mutations, respectively.** Top panel: The fraction (%) of aTn MEML within total AT-pair mutations (T➔A, T➔C, T➔G, and reverse compliments) are plotted on X-axis. The numbers on top of each bar notes the total aTn MEML count in respective cell types. Bottom panel: The fraction (%) of nCg MEML within total CG-pair mutations (C➔A, C➔T, C➔G, and reverse compliments) are plotted on X-axis. The numbers on top of each bar notes the total nCg MEML count in respective cell types. All source data are available in S5c Table in S5 Table.
(TIF)

**S8 Fig. Correlation analyses between aTn and nCg mutational motif enrichment and their corresponding sub-motif enrichment per donor. A.** aTn enrichment per donor are plotted against aTr enrichment per donor for indicated cell types to detect true aTn mutagenesis. Correlation coefficient and two-sided p value from Spearman's correlation

analyses are indicated on each plot. Correlation analysis was not conducted for the sun-exposed donors but was still displayed in the plot, as there were only two donors in that group. **B-C.** nCg enrichment per donor are plotted against **(B)** rCg enrichment per donor and **(C)** yCg enrichment per donor for indicated cell types to detect true nCg mutagenesis. Correlation coefficient and two-sided p value from Spearman's correlation analyses are indicated on each plot. Correlation analysis was not conducted for the sun-exposed donors but was still displayed in the plot, as there were only two donors in that group. Axis ranges for scatter plots were scaled to the specific range of data points within each cell type in order to maximize the resolution of internal trends and data distributions. All source data and p values from statistical analyses are available in S5 Table.
(TIF)

**S9 Fig. Decomposition of mutational motif enrichment.** The two components of calculating enrichment per donor, mutation density per motif and mutation density in genomic background, are plotted on log10 scale for (A) nCg, (B) rCg, and (C) yCg motifs. Each dot represents a donor. Size represents enrichment values. Colors indicate cell type. Ellipses show 95% confidence intervals. All source data are available in S5e Table in S5 Table.
(TIF)

**S10 Fig. Motif-motif and motif-age correlation analyses. A-B.** yTn➔yGn MEML are plotted against **(A)** yCn➔yTn MEML and **(B)** nTt➔nCt MEML for skin fibroblast samples. **C-E.** Average donor MEML for **(C)** aTn and **(D)** nCg motifs are plotted against donor age for both chondrocyte clone and single cell cohorts. **(E)** aTn and nCg motif MEML are plotted against donor age for down-sampled single chondrocyte cohort. Correlation coefficient and two-sided p value from Spearman's correlation analyses are indicated on each plot. Red line indicates best-fit linear regression. Axis ranges for scatter plots were scaled to the specific range of data points within each cell type in order to maximize the resolution of internal trends and data distributions. All source data and p values from statistical analyses are available in S5 Table.
(TIF)

**S11 Fig. Distribution of allele fraction of indels detected in chondrocyte samples sequenced in this study.** Allele fractions of all indels before filtering are plotted in bins of five. Samples originating from the same donor are indicated by color.
(TIF)

**S12 Fig. InDels in different chondrocyte types. A.** Average count per donor was shown in boxplots for total indels, indels in homonucleotide runs, and deletions ≥5 bp in different chondrocyte types stratified by type of DNA sequenced. Each dot represents an individual donor. Asterisks represent statistical significance (Wilcoxon Rank Sum test) between connected cell types. Black solid connector lines indicate two-sided test, red dashed connector lines indicate one-sided test. *P value ≤ 0.05. **B.** Average counts per donors plotted against donor age for total indels (top panel), indels in homonucleotide runs (middle panel), and deletions ≥5 bp (bottom panel) within each chondrocyte type indicated. Correlation coefficient and one-sided p value from Spearman's correlation analyses are indicated on each plot. Axis ranges for scatter plots scaled to the specific variance of data points within each cell type in order to maximize the resolution of internal trends and data distributions. For direct quantitative comparisons between groups using standardized scales, refer to the corresponding box plots in S12A Fig. All source data and p values from statistical analyses are available in S6 Table.
(TIF)

**S13 Fig. Cosine similarity and relative contribution of published ID signatures. A.** ID83 profiles were generated from pooled indels of indicated cell types and fitted with published COSMIC ID signatures to determine their relative contributions. **B.** Cosine similarity was calculated between the ID83 profiles of indicated cell types and published ID signatures. All source data are available in S6 Table.
(TIF)

**S1 Table. Demographic information and coverage statistics of chondrocyte samples sequenced in this study.**
(XLSX)

**S2 Table. Analyses of spectrum and accumulation rate of single base substitutions.** Datasheets in S2 Table: S2a Table: Mutation load in individual chondrocyte and fibroblast genomes. S2b Table: Mean and median total substitution counts and mutation accumulation rates in different cell types with corresponding P- values from Wilcox rank sum test. S2c Table: Spearman correlation analyses between average substitution load per donor and donor age stratified by cell type. S2d Table: Calculation of rate of mutation accumulation in individual chondrocyte and fibroblast clone genomes. S2e Table: Ratio of different base substitutions on transcribed and untranscribed strands of indicated cell types including p values from two-sided Poisson test. S2f Table: Count of mutations in chondrocyte clones and single cells that are analyzed, along with counts that are detected in clones and estimated in single cells. S2g Table: Linear mixed-effects (LME) model summary of mutation burden in chondrocyte clones and single cells.
(XLSX)

**S3 Table. Analyses of published SBS signatures.** Datasheets in S3 Table: S3a Table: Absolute contribution of published COSMIC SBS signatures in the mutation catalogues of chondrocytes and fibroblasts. S3b Table: Cosine similarity of COSMIC SBS signatures with trinucleotide mutation profiles of chondrocyte and fibroblast genomes.
(XLSX)

**S4 Table. List of mutational motifs analyzed in this study.**
(DOCX)

**S5 Table. Analyses of mutational motifs.** Datasheets in S5 Table: S5a Table: Percentage of donors with average MEML value >0 for indicated motifs and cell types. S5b Table: Mean and median motif enrichment and MEML in different cell types for indicated motifs with corresponding P- values from Wilcox rank sum test. S5c Table: Contribution of aTn and nCg MEML respectively to all AT and CG pair mutations detected in different cell types. S5d Table: Motif-motif correlation analyses. S5e Table: Decomposition of mutational motif enrichment per donor into component mutation rates. S5f Table: Spearman correlation analysis between donor age and average donor MEML for indicated motifs and cell types. S5g Table: Sample-specific enrichment and MEML values for all motifs and samples analyzed in this study.
(XLSX)

**S6 Table. Analyses of small InDels.** Datasheets in S6 Table: S6a Table: Counts of different indel types detected in chondrocyte and fibroblast samples; S6b Table: Mean and median of different InDels classes in different cell types with corresponding P- values from Wilcox rank sum test. S6c Table: Spearman correlation analyses between different indel classes and donor age stratified by cell type. S6d Table: Absolute contribution of known COSMIC ID signatures in the indel matrices of chondrocytes and fibroblasts. S6e Table: Cosine similarity of known COSMIC ID signatures with the indel matrices of chondrocytes and fibroblasts. S6f Table: Relative contribution of different types of InDels as determined by SigProfileMatrixGenerator to the total indel load in the chondrocyte and fibroblast cohorts.
(XLSX)

**S7 Table. Functional annotation of chondrocyte variants.** Datasheets in S7 Table: S7a Table: All SNVs in InDels in chondrocyte clones and single cells are annotated for functional effect and oncogenic prediction using ANNOVAR and CGI, respectively. S7b Table: List of OA effector genes obtained from GWAS meta-analysis performed in Hatzikotoulas *et al.* 2025. S7c Table: List of 2,351 mutations in chondrocyte clones and single cells that are on OA-associated genes.
(XLSX)

**S8 Table. Analyses of large structural variants.** Datasheets in S8 Table: S8a Table: List of all structural variants detected in chondrocyte and fibroblast genomes. S8b Table: Average number of structural variants detected in a donor,

stratified by cell and variant type. S8c Table: P-values from Wilcox rank sum test between average number of rearrangements detected in donors within different cell types. S8d Table: Total number of rearrangements overlapping with CFS and hotspots in each cell type.
(XLSX)

## Acknowledgments

The contributions of the NIH authors are considered Works of the United States Government. The findings and conclusions presented in this paper are those of the authors and do not necessarily reflect the views of the NIH or the U.S. Department of Health and Human Services. We thank the families of tissue donors to the Gift of Hope Organ & Tissue Donor Network, and Rush University and Mrs. Arnavaz Hakimiyan for tissue procurement. We also recognize the Department of Orthopaedics at the University of North Carolina at Chapel Hill for sample collection. We thank Drs. Natalya Degtyareva and Drs Rajula Alleva-Elango for advice on manuscript, and Mr. Adam Burkholder for help with data management.

## Author contributions

**Conceptualization:** Safia Mahabub Sauty, Brian O Diekman, Dmitry A. Gordenin.

**Data curation:** Safia Mahabub Sauty, Hamed Bostan, Piotr A Mieczkowski.

**Formal analysis:** Safia Mahabub Sauty, Hamed Bostan.

**Funding acquisition:** Brian O Diekman.

**Investigation:** Safia Mahabub Sauty, Jacqueline Shine, Piotr A Mieczkowski.

**Methodology:** Jacqueline Shine.

**Resources:** Richard F Loeser.

**Supervision:** Jian-Liang Li, Brian O Diekman, Dmitry A. Gordenin.

**Visualization:** Safia Mahabub Sauty.

**Writing – original draft:** Safia Mahabub Sauty, Dmitry A. Gordenin.

**Writing – review & editing:** Safia Mahabub Sauty, Jacqueline Shine, Richard F Loeser, Brian O Diekman, Dmitry A. Gordenin.

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
