## [Decision Letter · Decision Letter 0]

31 Mar 2026

PGENETICS-D-26-00135

Comparative whole-genome analyses of articular chondrocytes and skin fibroblasts reveal distinct genome instability landscapes in mesenchymal cell types

PLOS Genetics

Dear Dr. Gordenin,

Thank you for submitting your manuscript to PLOS Genetics. After careful consideration, we feel that it has merit but does not fully meet PLOS Genetics's publication criteria as it currently stands. All three expert reviewers agree that your manuscript has potential to make a strong contribution to this field, and they offered valuable feedback, requests and suggestions that will further improve its impact. Therefore, we invite you to submit a revised version of the manuscript that addresses the points raised during the review process.

We look forward to receiving your revised manuscript.

Kind regards,

Juan Lucas Argueso

Guest Editor

PLOS Genetics

Monica Colaiácovo

Section Editor

PLOS Genetics

Aimée Dudley

Editor-in-Chief

PLOS Genetics

Anne Goriely

Editor-in-Chief

PLOS Genetics

**Journal Requirements:**

At this stage, the following Authors/Authors require contributions: Dmitry A. Gordenin. Please ensure that the full contributions of each author are acknowledged in the "Add/Edit/Remove Authors" section of our submission form.

The list of CRediT author contributions may be found here: https://journals.plos.org/plosgenetics/s/authorship#loc-author-contributions

3) In the online submission form, you indicated that your data will be submitted to a repository upon acceptance. We strongly recommend all authors deposit their data before acceptance, as the process can be lengthy and hold up publication timelines. Please note that, though access restrictions are acceptable now, your entire minimal dataset will need to be made freely accessible if your manuscript is accepted for publication. This policy applies to all data except where public deposition would breach compliance with the protocol approved by your research ethics board. If you are unable to adhere to our open data policy, please kindly revise your statement to explain your reasoning and we will seek the editor's input on an exemption.

4) Some material included in your submission may be copyrighted. According to PLOS's copyright policy, authors who use figures or other material (e.g., graphics, clipart, maps) from another author or copyright holder must demonstrate or obtain permission to publish this material under the Creative Commons Attribution 4.0 International (CC BY 4.0) License used by PLOS journals. Please closely review the details of PLOS's copyright requirements here: PLOS Licenses and Copyright. If you need to request permissions from a copyright holder, you may use the PLOS Content Copyright Permission Form Please respond directly to this email and provide any known details concerning your material's license terms and permissions required for reuse, even if you have not yet obtained copyright permissions or are unsure of your material's copyright compatibility. Once you have responded and addressed all other outstanding technical requirements, you may resubmit your manuscript within Editorial Manager. Potential Copyright Issues:

i) Figure S1. Please confirm whether you drew the images / clip-art within the figure panels by hand. If you did not draw the images, please provide (a) a link to the source of the images or icons and their license / terms of use; or (b) written permission from the copyright holder to publish the images or icons under our CC BY 4.0 license. Alternatively, you may replace the images with open source alternatives. See these open source resources you may use to replace images / clip-art: - https://commons.wikimedia.org - https://openclipart.org/.

**Reviewers' comments:**

Reviewer's Responses to Questions

**Comments to the Authors:**

Reviewer #1: This manuscript by Sauty et al., conducted whole-genome sequencing of human chondrocytes and compared the mutational landscape with skin fibroblasts. Mutational signature and motif-centered analyses indicate that chondrocytes mainly include endogenously accumulated mutations associated with aging. In comparison, skin fibroblasts show strong mutational patterns induced by UV radiation. Overall, this is an interesting report comparing somatic mutations between two types of skin cells. The data interpretation is clear and the underlying mechanisms are discussed reasonably well.

My only comment is related to chondrocytes collected from donors with osteoarthritis (OA). The authors mentioned there is little difference between healthy chondrocytes and OA-chondrocytes. But they may have missed an opportunity to identify mutation sites in important genes or gene regulatory elements associated with OA. Adding additional mutation analysis to identify potential OA-associated gene mutations may increase the significance of this study.

Reviewer #2: In the presented manuscript, Sauty and colleagues describe their study of mutations in chondrocytes. The authors utilize cloning of single chondrocytes to investigate mutations in individual cells from nine donors. Based on the resulting mutation sets, they analyze mutation burden and spectra, mutational motifs, and structural variations. To expand the dataset, they also incorporate mutations identified in prior single-cell studies of chondrocytes from more than a dozen additional donors (ref. 21).

Several donors with osteoarthritis (OA) were included both in this and in the previous study. Somewhat surprisingly, both studies report minimal differences between mutational patterns in chondrocytes derived from OA versus normal donors.

The authors further compare the mutational landscapes of chondrocytes and skin fibroblasts. Although both cell types originate from the mesoderm, they reside in distinct anatomical environments and are subject to different endogenous processes and environmental exposures. Consistent with this, the authors identify multiple differences between fibroblasts and chondrocytes, including:

* Distinct mutational burdens and spectra for both SNVs and indels

* Greater influence of UV-induced damage in fibroblasts

* Stronger contribution of endogenous mutagenesis in chondrocytes

* Differences in the distribution of structural variations across genomic regions

The study leverages both signature-based and motif-based analyses of mutations, which make the analyses well rounded. I would like to emphasize this as a particular strength of the work. Mutational signature analysis is a broadly applicable approach that can identify patterns reflecting underlying mutational processes without prior assumptions. However, it operates in a 96-dimensional space (96 features), and the large (more than 96) number of known signatures complicate decomposition, leading either to contamination by multiple signatures or failure to detect biologically relevant ones.

In contrast, motif-based analysis is less general—since it relies on prior knowledge of sequence context—but offers greater robustness in quantifying contributions from known mutational processes. This complementarity is well illustrated in both the authors’ previous work and the current study. For example, signature-based analysis detected multiple signatures across clones and cells, yet SBS1 was identified only in chondrocyte clones and was missed in other datasets (Figure 2A). Given that SBS1 reflects spontaneous deamination of methylated cytosines and is expected to be ubiquitous across human cell types, this result is somewhat unexpected. Motif-based analysis, however, confirms its presence across datasets (Figure 3). Notably, the previous study of chondrocytes (ref. 21) also failed to detect SBS1 using signature-based approaches.

These observations underscore how the two analytical frameworks complement each other, and it is commendable that the authors applied both. Importantly, motif-based analysis also enabled a novel insight: chondrocytes appear to be more strongly influenced by endogenous mutagenic processes than fibroblasts. The authors also appropriately address potential overlap between motifs associated with different mutational processes.

Overall, I find the analyses presented in this manuscript compelling. However, I would like the authors to address the following points prior to publication:

* In general, comparisons of mutational burden (SNVs, indels, and SVs) between single cells and clones are not particularly informative. While it is reasonable to expect that fibroblasts from sun-exposed skin carry higher mutation loads than chondrocytes, more subtle differences—such as those shown in Figure 6A between cloned and single chondrocytes—may be influenced by technical artifacts. The experimental approaches and variant-calling methods differ substantially between single-cell and clonal datasets. I suggest moving such comparisons to the Supplementary Materials.

* Figure S2: Given that mutations were selected within a narrow VAF range (45% < VAF < 55%), mutation burden estimates for clones A06, A08, and A11 may be substantially biased because of peak in VAF distribution is not at 50%. The authors should comment on the potential biased mutation calling for those clones or consider removing them from the analyses.

* Lines 283–285: The term “subclonal nature” is unclear in this context and should be explicitly defined.

* Lines 363–365: The definition of the enrichment metric is difficult to follow. It would be helpful to move this explanation to the Methods section and/or provide a schematic, equation, or supplementary figure for clarity.

* Across multiple figures, axis ranges (e.g., for age, enrichment, and mutation burden) vary substantially, making comparisons between panels difficult. For example, in Figure 1B, the y-axis for chondrocytes ranges from 100–650 in one panel and 200–800 in another, while the x-axis ranges from 55–90 and 30–90. Such inconsistencies hinder interpretation. I recommend standardizing axis scales within panels for the following figures: 1B, 5A–C, 6B–D, 7B, S3B, S7A–C, and S10B.

Reviewer #3: Sauty et al. sequenced single-cell-derived chondrocyte colonies from healthy individuals and those suffering from osteoarthritis and provide analysis of somatic base substitution, indel and structural variants. Their data were supplemented by previously published whole genome-amplified single chondrocytes - which served a confirmatory role - and skin fibroblasts - which served as a contrasting mesodermal cell type. These data address questions about what mutagenic processes are active in healthy and diseased human tissues, an area of active research that has accelerated in recent years due to technological advances. The authors' study is both a timely and valuable contribution to these ongoing efforts.

Comparison of the three datasets revealed no significant difference between healthy and osteoarthritic chondrocytes. It did, however, point toward differences in the types and rates of mutations accumulated by chondrocytes and fibroblasts, and these differences were consistent with the proliferative capacities and exposures to exogenous mutagens of the two cell types. The authors' preferred approach, motif-centered analysis, differs from the more widely-used signature analysis and allows some mutagenic processes to be more directly probed.

While most of the conclusions are very sensible, some are troubling. For example, none of the motifs related to endogenous mutagenesis were correlated with donor age in clonally expanded chondrocytes, despite the main message of the work being that these same endogenous processes are the active ones in these cell types. Instead, this claim was substantiated only by the previously published single-cell amplified chondrocytes. I suspect this may be related to a lack of normalization between the 3 datasets compared in this study, which is itself a considerable challenge due to the very different technical approaches taken. However, these concerns can be reasonably addressed through a (hopefully not too burdensome) few changes to the analysis.

Major concerns

1. The authors compare their own single cell colony mutation counts to those from a single-cell study and find similar counts (Figure 1A-B). I find this surprising since there appears to be no correction for sensitivity for either the authors' own data or the SCMDA data. Given the technological differences I would expect sensitivity to be fairly different, especially given that SCMDA's sensitivity has been estimated between 40-50% (PMID 31467286).

2. The finding that healthy chondrocytes had higher burdens than OA chondrocytes is surprising. However, to make this claim the authors must account for differences in age distributions between the OA and healthy cohorts. This could be done by performing statistics on the residuals from the linear regressions against age. This would also benefit from correction for sensitivity since sensitivity between single cells can be quite variable due to the nature of the amplification.

3. Continuing the sensitivity point: another reason the authors should correct for sensitivity is so their aging rates for chondrocytes are compatible with the growing literature on aging rates. Their reported counts of ~300 substitutions for a 50-year-old is far below expectation, with even non-proliferating cell types gaining ~20 substitutions per year (=1000 substitutions at age 50, PMID 33911282). Correcting for sensitivity might also improve the linear regression fit.

4. It is unclear how accurately cell division count can be predicted by the telomere length estimates. The authors should provide a supplementary figure showing telomere length estimates vs. age to give a sense of how variable the estimates are. Additionally, SBS1 - one of the clock-like substitution signatures - is thought to be indicative of cell division. The authors could support their findings by showing a correlation between SBS1 burden and their telomere-based division estimates.

5. How do the authors explain that the hTg motif was found in all single cell chondrocyte categories but none of the clonally expanded chondrocytes? Could it be a single-cell sequencing artifact?

6. Is there really a lack of correlation (the P-value is 0.036) between nCg MEML and rCg enrichment in fibroblasts (Figure 5B?). There seems to be a very nice linear relationship between almost all points except 2-3 at the bottom right. This would seem to be consistent with the authors' interpretation since rCg does not overlap SBS7b (yCg).

7. Line 379-380: am I understanding correctly that the authors are stating the rightmost 2 bars in the top right of Figure 4A are similar? Perhaps the statistics are not significant when comparing the groups, but that likely is only reflecting the n=2 sun-exposed fibroblasts rather than evidence that the enrichments are similar.

8. It is very concerning that motif-age correlations were found only in the previously published single-cell chrondrocyte data and not supported by the authors' own clonally expanded chondrocytes. The explanation for this (lines 428-430) would be easier to accept if some more analysis was shown - e.g., at least a supplementary figure with line plots showing why the correlation coefficients are insignificant for the motifs in clonally expanded chondrocytes.

9. I was very surprised to find after reading the main text that the data were a mixture of two very different sequencing platforms: NovaSeq and DNBSEQ. Different sequencing technologies have different biases, and these artifact modes are often specific to exactly the sort of trinucleotide motifs analyzed in this work. The authors should (1) provide a table describing which technology was used for which samples (apologies if there is already such a table, I may have overlooked it) and (2) provide some analysis showing that NovaSeq and DNBSEQ provide similar signatures/motif enrichments. (2) need not be an exhaustive re-analysis of everything in the manuscript, but some reasonable confidence must be built.

Minor comments

1. The authors might consider using the standard colors for C>A (blue), C>G (black), C>T (red), etc. base changes as popularized by COSMIC. This could make it easier for readers familiar with the literature.

2. This is not the topic of this work, so only a minor point: the authors show a large increase in T>G, in sun-exposed fibroblasts, yet UV light exposure signatures feature C>T, T>A and T>C (signatures SBS7a-b, SBS7c and SBS7d from COSMIC, respectively). The authors comment that the T>Gs are hard to assign to a mutagenic source, so what could explain them? They don't seem to be technical artifacts since they do not occur in the new clonally expanded chondrocytes.

3. Related to (2), the axis break in Figure 1A completely hides the T>A burdens in sun-exposed fibroblasts and makes it hard to tell how much T>C and C>T there is. Perhaps the authors could make this clearer?

4. It would be helpful to present the substitution spectra for the 7 cell type/technologies in Figure 2A.

5. Line 478: what are "homonucleotide runs of 1bp repeat units"? Does this mean a single base deletion not in a homopolymer?

**Have all data underlying the figures and results presented in the manuscript been provided?**

Reviewer #1: Yes

Reviewer #2: Yes

Reviewer #3: **No:** The data are protected so I cannot access them, but this is typical for whole genome human data. I responded no only to state that I cannot verify whether the data have been deposited.

PLOS authors have the option to publish the peer review history of their article (what does this mean?). If published, this will include your full peer review and any attached files.

**Do you want your identity to be public for this peer review?** For information about this choice, including consent withdrawal, please see our Privacy Policy.

Reviewer #1: **Yes:** Peng Mao

Reviewer #2: No

Reviewer #3: No

**Figure resubmission:**
---

## [Decision Letter · Decision Letter 1]

6 May 2026

Dear Dr Gordenin,

We are pleased to inform you that your manuscript entitled "Comparative whole-genome analyses of articular chondrocytes and skin fibroblasts reveal distinct genome instability landscapes in mesenchymal cell types" has been editorially accepted for publication in PLOS Genetics. Congratulations!

Yours sincerely,

Juan Lucas Argueso

Guest Editor

PLOS Genetics

Monica Colaiácovo

Section Editor

PLOS Genetics

Aimée Dudley

Editor-in-Chief

PLOS Genetics

Anne Goriely

Editor-in-Chief

PLOS Genetics

BlueSky: @plos.bsky.social

Comments from the reviewers (if applicable):

Reviewer's Responses to Questions

**Comments to the Authors:**

Reviewer #1: The authors have addressed my main suggestion with more data analysis. I support publication of the revised manuscript.

Reviewer #2: The authors adequately address my concerns.

**Have all data underlying the figures and results presented in the manuscript been provided?**

Reviewer #1: Yes

Reviewer #2: Yes

PLOS authors have the option to publish the peer review history of their article (what does this mean?). If published, this will include your full peer review and any attached files.

Reviewer #1: No

Reviewer #2: No

**Data Deposition**

http://datadryad.org/submit?journalID=pgenetics&manu=PGENETICS-D-26-00135R1

**Press Queries**

---

## [Editor Report · Acceptance letter]

PGENETICS-D-26-00135R1

Comparative whole-genome analyses of articular chondrocytes and skin fibroblasts reveal distinct genome instability landscapes in mesenchymal cell types

Dear Dr Gordenin,

We are pleased to inform you that your manuscript entitled "Comparative whole-genome analyses of articular chondrocytes and skin fibroblasts reveal distinct genome instability landscapes in mesenchymal cell types" has been formally accepted for publication in PLOS Genetics! Your manuscript is now with our production department and you will be notified of the publication date in due course.

With kind regards,

Anita Estes

PLOS Genetics

On behalf of:
